# Science, interrupted: Funding delays reduce research activity but having more grants helps

**Wei Yang Tham** [ID] *

Cotting House, 50 N Harvard St, Boston, MA, United States of America

* wtham@hbs.edu

**Data Availability Statement:** The key data in this paper on grant transactions are confidential and housed at the Institute for Research on Innovation and Science (IRIS) at the University of Michigan. Readers can apply for access to the data and code at https://iris.isr.umich.edu/research-data/access/

## Abstract

I study how scientists respond to interruptions in the flow of their research funding, focusing on research grants at the National Institutes of Health (NIH), which awards multi-year, renewable grants. However, there can be delays during the renewal process. Over a period beginning three months before and ending one year after these delays, I find that interrupted labs reduce overall spending by 50% but over 90% in the month with the largest decrease. This change in spending is mostly driven by a decrease in payments to employees that is partially mitigated when scientists have other grants to draw on.

## Introduction

In many fields of science, research is a resource-intensive endeavour. It requires people, capital, and the management of these resources. In addition, scientists must obtain and manage the funding necessary to acquire these inputs. This includes dealing with the possibility that funding may not arrive in the amount or at the time they want it to. How scientists respond to uncertainty and liquidity constraints in funding is therefore an important part of the research production function.

In general, however, this aspect of a scientist's job is difficult to observe on a large scale. The UMETRICS dataset [1], which consists of administrative data from universities on transactions from sponsored projects, helps to bridge this gap. I use UMETRICS to study how Principal Investigators (abbreviated as PIs; for the remainder of the paper, I use the terms "lab" and "PI" interchangeably) funded by the National Institutes of Health (NIH) respond to funding delays or "interruptions".

I first document that when funding is guaranteed and available, scientists tend to maintain spending at a steady level (Fig 1). On average, after a "ramping up" period in the first year of a *project period* (i.e. the NIH term for a multi-year grant), spending is relatively flat until the final year of the project period, when it steadily decreases. This pattern suggests that in the absence of uncertainty about funding or liquidity constraints (and conditional on how the NIH disburses funds), scientists have a preference for a stable rate of spending.[1]

Next, I study how scientists respond to funding delays or "interruptions", focusing on a particular type of NIH grant, the "R01". R01 grants are generally regarded as being necessary to establish an independent research lab in the biomedical sciences. They can be renewed

or contact IRISdatarequests@umich.edu for guidance. Data that can be shared are hosted on the Open Science Framework at https://osf.io/ekq47/ (DOI: 10.17605/OSF.IO/EKQ47) Citation data were obtained under license from Clarivate Analytics (https://clarivate.com). Readers can contact Jeffrey Clovis (IP&Science) jeff.clovis@Clarivate.com and Ann Beynon (IP&Science) ann.kushmerick@Clarivate.com for information on obtaining the same data.

**Funding:** WYT received support from the National Institute on Aging of the National Institutes of Health under Award Number R24AG048059 to the National Bureau of Economic Research https://www.nia.nih.gov/ The funders had no role in study design, data collection and analysis, decision to publish, or preparation of the manuscript.

**Competing interests:** The authors have declared that no competing interests exist.

periodically (typically every four to five years) at the end of each project period, at which point the following scenarios may occur:

1. The project's new funding stream begins as soon as its previous one ends.

2. The project is *interrupted*: its new funding stream only begins some time after its previous one ends.[2]

Interruptions can arise for different reasons. A funding agency that is uncertain about its budget may engage in "precautionary saving" and delay spending to the end of the fiscal year [2]. There may also be disruptions to the funding allocation process, such as the government shutdown of Fiscal Year 1996 [3], which slowed down the processing of paperwork and led to peer review meetings being postponed (see also Appendix Section).

An interruption can be thought of as a combination of (a) a *liquidity* shock from the scientist's inability to access funding for some period of time and (b) an *uncertainty* shock about when or even whether they will get the funding in the first place. Although I do not distinguish between these two mechanisms, their combined effect in the form of interruptions is an important policy question. Delays in NIH funding are a real concern among researchers [4]. Understanding the role of interruptions as a potential impediment to science helps policymakers to determine how much attention should be paid to this issue.

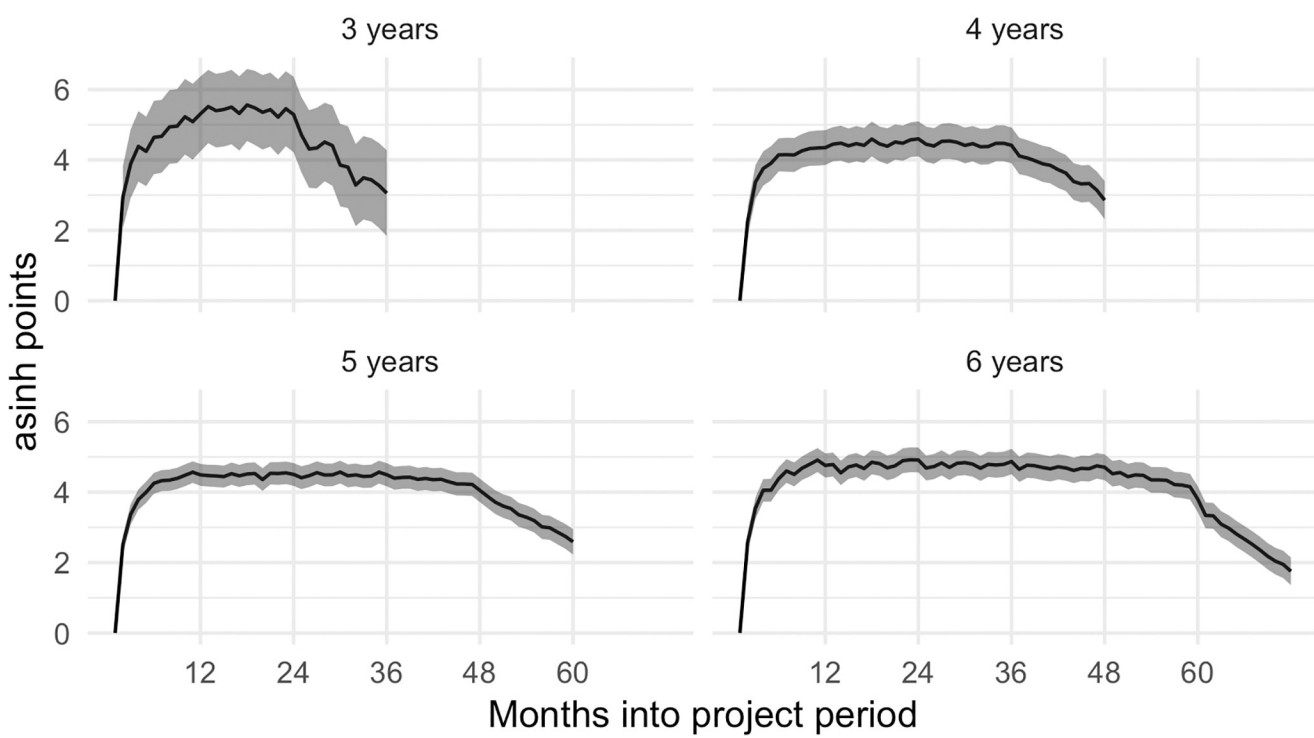

**Fig 1. This figure shows spending per month for R01 project periods that last three, four, five, or six years, relative to spending in the first month of the project period.** Estimates are from a regression of total expenditure (arcsinh-transformed) on a set of dummies for each month in a project period with project period fixed effects, with the first month of the project period as the excluded category. Separate regressions are run by project period length. Standard errors are clustered by expiring R01 project period.

I estimate the effect of interruptions with a difference-in-differences design that compares outcomes for interrupted and uninterrupted labs, defining "interrupted" projects as those where funding was renewed after more than 30 days.[3] To allow for the possibility that principal investigators (PIs) can dampen the effects of interruptions with other grants, I run the analysis separately on PIs with one R01 and PIs with multiple R01s.[4]

I find that interrupted labs with one R01 reduce spending significantly. Over a two-year period centered on the expiry of an R01 project period, interrupted labs spend 33% less than uninterrupted labs in the average month.[5] The change in spending is not uniform over time but V-shaped.[6] At its lowest point, spending is 96% lower for interrupted labs. This decrease starts about three months before the official grant expiry date and may be driven by uninterrupted labs spending from their renewed budget in advance, if they are informed of their successful renewal early enough. After R01 expiry, spending drops sharply before starting to recover, taking about nine months after expiry to catch up with uninterrupted labs. Over a 16-month period (from three months before expiry—when spending begins to decrease—to 12 months after grant expiry), the decrease in total spending is 52%.

When a PI has multiple R01s, spending remains stable throughout for interrupted PIs, indicating that there is fungibility across research grants. This is supported by the employee-level analysis. Across occupations, employees who are linked to multiple R01s either experience a lesser or zero decline in the probability of being paid by a grant, whether by their PI or by any grants at all.

I also look at whether PIs adjust different components of spending differently in response to interruptions. For PIs with one R01, both vendor and labor spending decrease substantially (by over 90% at their lowest points), but vendor spending does not recover as quickly. For PIs with multiple R01s, there is some decrease in the number of employees but vendor payments are relatively stable. The decrease in employees for both PI types may be due to labor expenditure constituting a larger share of spending and entailing longer-term commitments that PIs are unable to commit to until they know their funding status. However, this does not necessarily mean that employees are being dismissed by their institution or even removed from the research team, as there may be alternative sources of funding for some employees (e.g. teaching positions for graduate students), although some occupations (e.g. postdoctoral researchers) may be more vulnerable than others.

In my final set of results, I estimate the impact of a funding interruption on research output as measured by publications and citation-weighted publications. However, these estimates are not precise enough to determine whether interruptions affect output and, if they do, to what extent. This illustrates how traditional measures such as publications and patents may not provide the complete picture because they occur at a lower frequency and only capture one aspect of the research production process.

One limitation of this paper is that interruptions are not randomly assigned. While interruptions are driven in part by external events such as government shutdowns, the NIH may prioritize projects or PIs that are perceived to be of higher quality. This is less likely to be a problem for the results where inputs are an outcome, given that the use of inputs is more likely to be driven by budget constraints and the specific needs of the project. This is more likely to bias upward (in magnitude) the results involving publications, although the publication pre-trends do not indicate that interrupted PIs were on a less productive trajectory leading up to the year they were interrupted.

Another limitation is that I do not observe the scientists' full set of funds. UMETRICS does not record the spending of internal funds (i.e. those directly provided to a scientist by their institution). I interviewed a university administrator who works on grant management about the role of internal funding in such situations. While they could not put a number on the

extent to which internal funding makes up for the funding gap, they said that it was generally harder to get internal funding for personnel than non-personnel expenditures. Thus even when internal funds are an option, interruptions may still result in PI-employee separations.

In addition, my measure of PI spending is limited to spending through NIH grants. While this ensures a high degree of accuracy in linking NIH PIs to transactions, it also naturally raises the question of whether the amount of funding from non-NIH grants has a substantive effect on the results discussed thus far. This is unlikely for two reasons.

First, the results on whether interrupted employees continue to be paid on *any grant* are consistent with the overall set of results and thus do not suggest that there is a substantial pool of non-NIH grants being used to offset the effects of interruptions. Second, other research and government statistics show that if a research group is federally funded, it is also mostly federally funded and the NIH is the largest federal funder of life sciences research (see Appendix Section for more details). In short, focusing on NIH funding provides substantial coverage of researcher funding.

This paper builds on work using granular data to unpack the role of the "lab" in science, dating back to the anthropological work of [5] in 1979. On a larger scale, [6] use a complete personnel roster of principal investigators in the MIT Department of Biology from 1966 to 2000 to study the role of different types of personnel in research production. [7] use the UME-TRICS dataset as well to estimate the marginal product of scientific funding and show how employee composition changes when funding increases. This paper adds to the body of work by examining how uncertainty and liquidity constraints affect the use of research inputs and by highlighting the value of high frequency data in studying the knowledge production process.

This paper is also part of a literature in innovation economics studying how uncertainty affects innovators' productivity and choices. [8] study the Howard Hughes Medical Institute (HHMI) Investigator Program, which gives grantees more freedom over research direction and effectively gives them longer grant cycles compared to R01s, thus insulating them from the type of disruptions that can arise in the R01 renewal process. They find that HHMI scientists are more likely to produce high-impact papers and explore new research directions. While the insights from [8] are important, there are practical difficulties to expanding a resource-intensive program like the HHMI's. Thus, understanding where improvements can be made within the current system is important as well.

The results of this study highlight that a funding agency's decision to delay the renewal of a project may not be costless. Even when the project is eventually funded, there can be disruptions to the use of inputs, team capital [9], and the employment or training of personnel. This has two major implications for how we fund projects. Firstly, it suggests that there is value to having the budgets of science funding agencies planned over a longer-term horizon to reduce uncertainty [10]. Secondly, funding agencies delay projects if they expect that higher quality projects may be available later in the fiscal year. Agencies should consider that the cost of disrupting a project could be larger than the improvement in quality from delaying its decision, especially if their measures of project quality are imperfect.

## Background and conceptual framework

### NIH funding

The NIH is responsible for an annual budget of about US$40 billion, much of which is disbursed through research grants. A core part of the NIH's mission is funding basic science to generate fundamental knowledge that tends to have long-term, rather than immediate, impact.

The NIH is funded every fiscal year by congressional appropriation. This is part of a broader process whereby the US Congress passes regular appropriations bills to fund a wide range of government operations.[7] If appropriations have not been made by the beginning of the fiscal year, Congress can enact a "continuing resolution" to provide temporary funding. If a continuing resolution is not enacted and a "funding gap" occurs, then federal agencies have to begin a "shutdown" of projects and activities that rely on federal funds.

It is typically taken as given that regular appropriations will not have been made by the beginning of the fiscal year on 1 October, and that federal agencies will have to operate under a continuing resolution for at least some portion of the year. Under a continuing resolution, the NIH continues to fund existing projects, albeit at an inititally reduced rate. However, it might also choose to delay funding for new or renewed projects in response to uncertainty about the size of the NIH's budget for the fiscal year.

To illustrate, suppose that at the beginning of the fiscal year, the NIH knows (1) its budget and (2) its own ranking of projects available to be funded (rank could be based on project quality but also other factors such as NIH priorities). In this scenario, the NIH knows which projects it wishes to fund *and* whether it can fund them before the projects are set to run out of funding. Thus, there are no funding interruptions.

The scenario above illustrates that funding interruptions arise from uncertainty about either (1) the NIH's budget or (2) the quantity and quality of projects that need funding that fiscal year, or (3) both. Some uncertainty over projects is built-in as there are three review cycles throughout the fiscal year.

## Scientist perspective

The R01 is designed to provide enough funding to establish an independent research career. An R01 *project period* lasts for 4–5 years, after which it must be renewed in order to receive additional funding.[8] The same *project* can last for multiple *project periods*.

Ideally, a researcher wants to maintain R01 funding for as long as possible. Toward the end of each project period, the principal investigator (PI) has to apply to renew their project for another project period of 4–5 years. PIs typically apply for renewal 1–2 years before a project period ends in order to receive funding continuously. In addition to the time taken to prepare the application itself, PIs have to take into account other factors such as potentially having to resubmit an application that is rejected the first time.

## Data and variable construction[9]

### UMETRICS

I use the 2019 release of the UMETRICS data, which is housed at the Institute for Research on Innovation and Science (IRIS). UMETRICS core files are administrative records of transactions from sponsored projects from 31 member universities. The time span covered by each university's records varies, spanning 2001 to 2018 overall.[10] Payments from a project can go to one of three categories: vendors, subawards, or personnel.[11]

**Lab/PI total direct expenditure**. A key outcome variable is direct expenditure from grants, which excludes the overhead costs that are paid to universities as a percentage of a grant award. Although I define the timing and length of funding delays around the R01 grant, I sum up outcomes to the level of the PI/lab. Specifically, for each R01, I find its associated PIs at the point of renewal. I then sum up spending for each PI across all NIH grants that they are associated with at a given point in time.[12]

**Lab/PI vendor and labor expenditure**. I repeat the above procedure for payments to vendors and payments to labor. Payments to vendors includes purchases of equipment or services. UMETRICS does not include salaries, so payments to labor are backed out as the remainder after subtracting vendor and subaward payments from total expenditure (i.e. *Labor = Total − Vendor − Subaward*).

**Lab/PI employee counts**. Count of the number of employees paid by a PI through the PI's NIH grants.

*Employee-level outcomes*. The next part of my analysis is at the individual employee level. The UMETRICS data contains unique employee identifiers so that I can follow an employee's employment status over time. For each PI-R01-renewal combination, I identify employees paid by the PI *every month* over the 10–12 months before R01 expiry (i.e. the first three months of the panel). This is a heuristic to identify personnel who are more likely to be long-term members of the PI's lab or who were not already scheduled to end their tenure with the focal PI. I then create a monthly panel following their employment status from 9 months before expiry to 12 months after expiry.

I construct two outcome variables. The first is whether the employee is paid by the focal PI through any of the PI's grants in a given month. This can be thought of as a proxy for whether employee-PI matches are disrupted. The unit of interest is an employee-PI combination, and the data structure is an employee-PI-R01-renewal monthly panel.

The second outcome measure is whether the employee is paid by *any grants* from any PI in any given month. Even if an employee is separated from their usual PI they may be shifted to a different project, so this captures the overall "employment status" of the employee. The unit of interest here is an employee, so the data structure is an employee-R01-renewal monthly panel.

Both of these outcomes are non-absorbing—the indicator can be on-then-off or off-then-on in consecutive months and do not necessarily indicate that an employee as "exited" employment, which we do not observe.

UMETRICS also provides occupation categories for employees at the time they were paid. These classifications enable us to see how outcomes may evolve differently for different occupations. For example, we may expect the results for faculty to be different than the results for postdocs. For sample size reasons, any analysis that involves splitting the sample by occupation focuses on the largest five occupation categories: Faculty, Graduate Student, Post-graduate Researcher, Research, and Research Facilitation. A full list of occupational categories and descriptions is available in Appendix Section.

## ExPorter

ExPorter is publicly available data provided by the NIH.[13] It contains data on NIH-funded projects from 1985 to the present, including identifiers that IRIS has used to link projects to their transcations in UMETRICS. It also provides links to publications, patents, and clinical studies that cite support from the NIH. I describe the key variables constructed from ExPORTER below:

**Length of funding gaps and interruptions**. Number of calendar days between the end of a project period and the beginning of the next project period.

**Grant portfolio ("Number of R01s")**. An interruption's effect on a PI may vary by the size of their grant portfolio. I count the PI's number of NIH grants from one year before to one year after the focal R01 expired. I define the size of the PI's grant portfolio based on the number of "R01-equivalent" or "P01" grants that the PI had, including the focal R01. I follow the NIH definition of R01-equivalent grants.[14] P01 grants provide funding for multiple research

projects with a common theme.[15] For brevity, I will refer to this variable as the "Number of R01s" without explicitly defining the other types of grants included.

Since employees are not necessarily in charge of their own grants, I need to define funding support for employees. When the unit of interest is an employee-PI combination, this is the grant portfolio of the focal PI, as defined above. When the unit of interest is an employee, I identify all PIs that paid the employee at any time during the 10–12 months before R01 expiry. I then count the number of R01s (including R01-equivalents and P01s) that those PIs were in charge of during the 24-month period used in the analysis.

**Lab/PI publications**. I use publication counts as a proxy for research output. ExPorter provides a crosswalk between NIH projects and publications in PubMed, a database for publications in biomedical research and the life sciences. These can then be aggregated up to the PI or lab-level. I also weight publications by 3-year forward citations (i.e. citations from years $X$, $X + 1$, and $X + 2$ if a paper was published in year $X$).

### Author-ity & web of science

Author-ity [11] is a dataset of disambiguated author names based on a snapshot of MEDLINE, a bibliographic database with a focus on "biomedicine and health," in 2009.[16] Each observation in Author-ity is a cluster of names and articles that are predicted to belong to the same author. Clarivate Analytics Web of Science (WoS) is a citation indexing database that is not publicly available. I use a version of WoS that indexes articles and citations up to and including 2013.

The final sample for estimating the effect of interruptions on research output combines ExPORTER and WoS to create a panel that spans 1985 (earliest ExPORTER year) to 2011 (latest WoS year allowing for 3-year forward citations). Additional covariates such as career age are added from Author-ity. Section describes how this panel is constructed and how it relates to estimation.

### Empirical strategy

The goal of this study is to estimate how outcomes for labs/PIs change when they experience an R01 interruption. In doing so, it is important to take into account the decrease in spending observed at the end of a grant (Fig 1). Thus, a natural comparison group for labs with interrupted R01s is labs with R01s are at similar stages in the cycle (i.e. labs with R01s that were also expiring but were successfully renewed).

I begin by identifying instances where an R01 was successfully renewed within the fiscal year it expires. I then stack all combinations of renewed R01s and PIs of those R01s to create a balanced PI-R01-renewal monthly panel spanning 24 months—one year before and one year after the expiry. I define an R01 as (a) "interrupted" if took more than 30 calendar days to be renewed or (b) "uninterrupted" or "continuous" if it took fewer than 30 calendar days to be renewed. I then estimate event study specifications that allow us to see how labs respond to interruptions month-to-month.

### Event-study: Research inputs

The main specification I estimate is an event study centered around the expiry month of a project period.

$$y_{LRt} = \sum_{e=-10}^{12} \beta_e * \mathbf{1}(e = t - t_{expiry})\mathbf{1}(Interrupted) + \delta_{LR} + \gamma_e + \epsilon_{LRt}$$

I index PIs as $L$, R01s as $R$, and the year-month as $t$. $t_{expiry}$ is the year-month that the R01 grant $R$ expires. $e$ is the number of months before expiry (i.e. $e = 0$ when $R$ expires and $e < 0$ before the grant expires). I restrict the sample to the one year before and after the R01 $R$ expires, i.e. $e$ starts at month $-11$ and ends at 12. $e = -11$ is excluded from the specification.

$y_{LRt}$ is a variable at the PI-level, such as total spending across all of the PI's grants), $\delta_{LR}$ are PI-R01-renewal fixed effects, and $\gamma_e$ are fixed effects for months relative to expiry. The coefficients of interest are $\beta_e$, where $t = -10, -9, \ldots, 11, 12$.

An important literature on misspecification of the two-way fixed effects model for the staggered difference-in-differences design has emerged in recent years [12–15]. Compared to the typical setting considered in this literature, one advantage of the setting in this paper is being able to define a "post-treatment" period for control units, i.e., the expiry date of the R01 grant. This means that we can define time relative to treatment for both treatment and control units. Thus, the above specification estimates an event study for a difference-in-differences with only one treatment period *in relative time*, thereby avoiding the key issue raised in the literature.

**Employee-level outcomes.** Continuing with the same notation, I index employees as $i$. To assess whether the employee-PI relationship was affected, I estimate the following regression specification with employee-PI-R01-renewal fixed effects and relative time fixed effects:

$$\mathbf{1}(PI - paid - employee)_{iLRt} = \sum_{e=-10}^{12} \beta_e * \mathbf{1}(e = t - t_{expiry})\mathbf{1}(Interrupted) + \delta_{iLR} + \gamma_e + \epsilon_{iLRt}$$

For looking at the effects on the employee, I estimate:

$$\mathbf{1}(Any - grant - paid - employee)_{iRt} = \sum_{e=-10}^{12} \beta_e * \mathbf{1}(e = t - t_{expiry})\mathbf{1}(Interrupted) + \delta_{iR} + \gamma_e + \epsilon_{iRt}$$

## Event-study: Research outputs

Through ExPorter, I draw on the universe of NIH-sponsored scientists to estimate the effect of funding restrictions on research output. I use a "stacked regression" approach [16, 17].[17] For each calendar year, I find all R01-PI combinations where the R01 was eventually renewed. The set of all such R01-PI combinations for each year form a "treatment cohort". Each cohort is then stacked to form a PI-R01-renewal by year panel that starts 4 years before and ends 5 years after an interruption, restricted to all interruptions that took place from 1989 to 2006.[18]

I then estimate a variant of the two-way fixed effects specification but with cohort-specific time and unit fixed effects to address issues arising from staggered treatment timing in difference-in-differences [13, 15, 18, 19].

I index PIs with $i$, renewed R01s with $R$, and calendar year with $t$. $c$ indexes treatment cohorts or calendar year $R$ expired. Treatment cohorts are defined as the set of PIs that were renewed in the same calendar year, some of whom were interrupted and some not. $y_{iRt}$ is a measure of research production (e.g. publications), $\delta_{iR}$ are PI-R01-renewal fixed effects, and $\gamma_{tc}$ are treatment cohort by calendar year fixed effects. $t - t_{expiry}$ is years since interruption, with 0 being the interruption year.

I then apply the Coarsened Exact Matching (CEM) procedure [[20]; see the Appendix Section for details] estimate the following event-study specification and its "static" counterpart

with the matching weights:

$$y_{iRt} = \sum_{k=-5}^{5} \beta_k * \mathbf{1}(k = t - t_{expiry}) \mathbf{1}(Interrupted) + \delta_{iR} + \gamma_{tc} + \epsilon_{iRt}$$

$$y_{iRt} = \beta_{static} * \mathbf{1}(t - t_{expiry} \geq 1) \mathbf{1}(Interrupted) + \delta_{iR} + \gamma_{tc} + \epsilon_{iRt})$$

$$t - t_{expiry} = -4, -3, \ldots, 0, \ldots, 5$$

### Inverse hyperbolic sine

Unless otherwise stated or the outcome is a binary variable, I apply an inverse hyperbolic sine (also *asinh* or *arcsinh*) transformation to the outcome variable for all regressions, which approximates a natural logarithm and is defined at zero.[19]

### Descriptive statistics

The analysis within UMETRICS uses a sample of 356 PI-R01-renewals with one R01 (284 uninterrupted, 72 interrupted) and 693 PI-R01-renewals with multiple R01s (528 uninterrupted, 165 interrupted). The timing of R01 renewals ranges from 2003 to 2017, which results in a panel ranging from 2002 to 2018. Fig 11 in the Appendix shows the years in which R01 expiries in this sample occurred.

Fig 2 compares characteristics of the expiring (and eventually renewed) R01 project periods from the UMETRICS sample used in the upcoming analysis. Fig 2A and 2B show the distribution of funding gaps. Overall, a little over 20% of R01s were "interrupted" or renewed more than 30 days after their expiry date. The funding gap experienced by interrupted R01s has a wide range with a maximum of over 300 calendar days.

Fig 2C and 2D show that interrupted and continuous R01s are similarly funded, whether in terms of total funding over the entire project period or funding per year, with interrupted projects being slightly bigger. Interrupted projects are more likely to have had a six-year project period (Fig 2E).

R01s are awarded for a maximum of five years, thus six-year projects periods are likely to have originally been five-year awards that exercised a one-year no-cost extension and are more likely to be interrupted because they no longer have the option to extend. This might affect the results if spending trends differ by project length. I address this by exact matching on the project period length of the focal R01 for outcomes related to spending.[20]

In the first month of the panel, the median team for PIs with one R01 had 4 employees in total, 1 faculty, and 1 research employee, and 0 for the remaining occupations.[21] The median team for PIs with multiple R01s had 8 employees, 2 faculty, 1 research staff, 1 postgraduate researcher, and 0 for the remaining occupations. The most common occupations on the average team are faculty, research staff, graduate students, postgraduate researchers, and research facilitators, after which other occupations are much less likely to be on a team.[22] Median expenditure at the beginning of the panel for PIs with one R01 was $17,100 for total direct expenditure, $13,900 for labor payments, $900 for vendor payments, and 0 for subaward payments. For PIs with multiple R01s, the same statistics were $40,200 for total direct expenditure, $30,900 for labor payments, $4,100 for vendor payments, and $0 for subaward payments.

The analysis for research output (i.e. based on the ExPORTER database) uses a sample of 10890 PI-R01-renewals with one R01 (8934 uninterrupted, 1956 interrupted) and 11570 PI-R01-renewals with multiple R01s (9390 uninterrupted, 2180 interrupted). The timing of R01 renewals ranges from 2003 to 2017, which results in a panel ranging from 2002 to 2018. Fig 11 in the Appendix shows the years in which R01 expiries in this sample occurred.

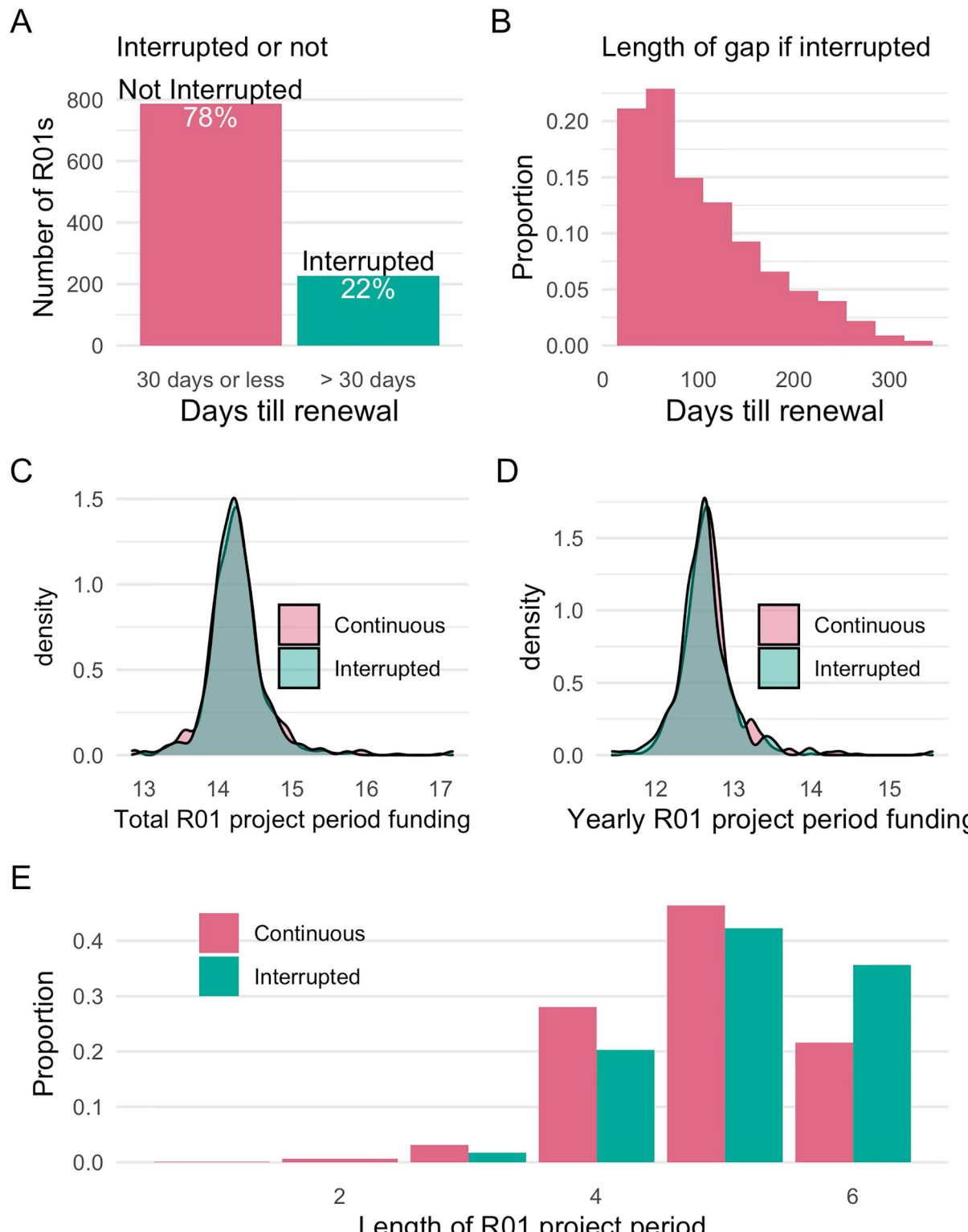

**Fig 2. This figure compares the characteristics of expiring (and eventually renewed) R01 project periods that are used in the UMETRICS analysis.** Each unit of observation is an R01 project period. Fig 1A shows the number of R01s that were renewed within 30 calendar days of their expiry. Fig 1B is a histogram (30-day bins) for the number of days till renewal for R01s not renewed within 30 days. Fig 1C shows the smoothed density for total funding in the expiring project period for interrupted and uninterrupted R01s. Fig 1D shows the same figure for funding per year (total funding / length of expiring R01 project period). Fig 1E shows the proportion of projects by the expiring project period's length.

## Results

### Spending

Fig 3A shows the event study estimates for total spending by PIs, with separate estimates by whether the PI had one R01 (green) or multiple R01s (brown) around the time of expiry. The "1 R01" graph (green) shows that for PIs with one R01, total spending starts decreasing about three months before the official expiry date. At the lowest point, spending is 96% lower for PIs with interrupted R01s. The decrease in the stock of spending from month -3 to month 12 is 52%.[23]

A priori, we might not expect interruptions to affect how PIs spend their funds if restrictions on when or how they can spend those funds make it difficult to deviate (e.g. if they can only spend those funds within the original budget period). Since we observe a divergence in spending even before the official expiry date, this indicates that PIs have some ability to shift spending outside of the official budget periods. This can happen in two ways.

First, PIs can engage in "pre-award spending", where they make a request to incur costs 90 days before the official start date of the grant.[24] Thus, if PIs of uninterrupted grants know early enough that they will be funded, some of the divergence in spending could be due to them making use of pre-award spending.

Second, PIs of interrupted grants may be be able to delay spending beyond the official end date of the grant. For example, they might want to hold off on hiring a postdoctoral researcher until there is more certainty that they have the funding to support them. While these actions are not mutually exclusive, a university administrator said that in their experience, pre-award spending was used "quite often", whereas saving funds for later was not common.[25] Based on these remarks and the timing of the divergence in spending, the first explanation (pre-award spending by uninterrupted grants) seems to be the more likely reason for what we observe.

Fig 3B and 3C show the event study estimates for labor payments and vendor payments respectively. For labor spending, the change in spending patterns look broadly similar to those for overall spending. Vendor payments decrease less than labor payments in *asinh* points but still substantially by percentage, with a 93% decrease at its lowest. Vendor payments also do not recover as quickly as labor payments.

Finally, Fig 3D shows the event study estimates using employee counts as the outcome variable. These results are consistent with what we see for labor payments. In the month after the expiry of the project period, interrupted PIs with one R01 pay about 87% fewer personnel. In the same month, interrupted PIs with multiple R01s pay about 22% fewer employees.[26]

### Employees

In addition to their effects on research production, interruptions may have disruptive effects on employees. One concern is that it may force employee turnover in a lab. For instance, a staff scientist may have to switch between PIs in order to maintain their salary or employment, or a postdoctoral researcher may be forced to leave if renewal funding does not become available quickly enough to fund their position. In addition to the personal disruption to employees, there may be a loss of team-specific capital [9].

To get at these issues, I focus on the following outcome variables: whether an employee was (a) paid by the same PI on an NIH grant or (b) paid on any grants at all. I subset the data by employee occupation and number of NIH grants an employee is associated with (as detailed in the *Employee-level outcomes* section).[27]

Fig 4 shows the event-study estimates by occupation subsample. Across all occupations, interrupted employees associated with one R01 are less likely to be paid by the same PI or by

## PI Response to Interruption

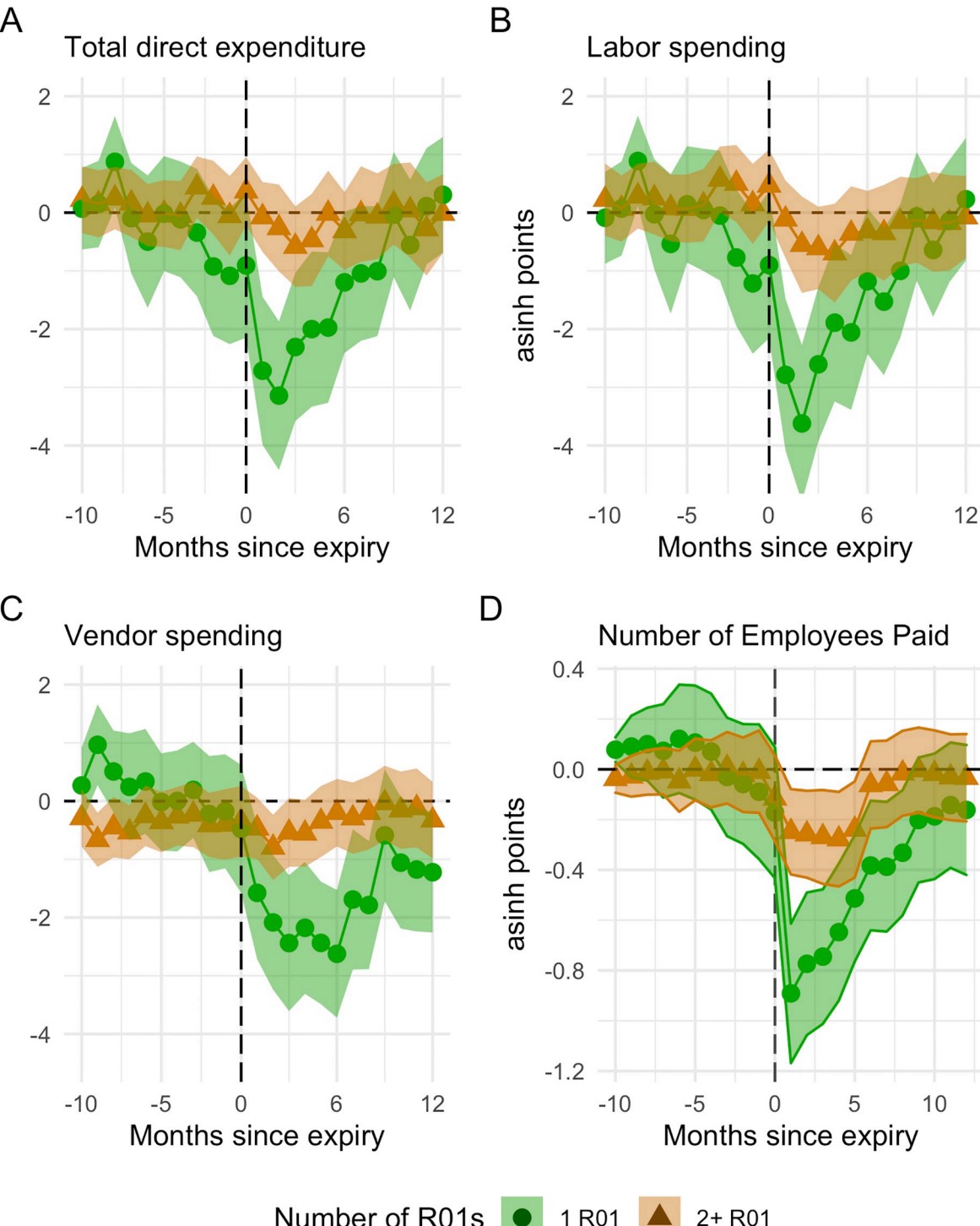

**Fig 3. This figure shows event-study estimates (with 95% confidence intervals) of the difference in spending between PIs of interrupted and uninterrupted R01s.** Treated and control groups are matched on length of the expiring R01 project period and the regression is weighted using the matching weights. Each panel shows the estimates for a different outcome variable: total expenditure by PI (A), total vendor expenditure by PI (B), total labor expenditure by PI (C), and total number of employees paid by PI (D). Month 0 is the month that the focal R01 expires. Month -11 is the excluded category for the regression. Regressions are run separately on subsamples of PIs that have one R01 grant (green) or multiple R01s (brown), including R01-equivalents and P01 grants. Standard errors are clustered by expiring R01 project period.

## Employee status when interrupted

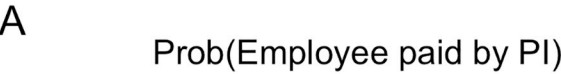

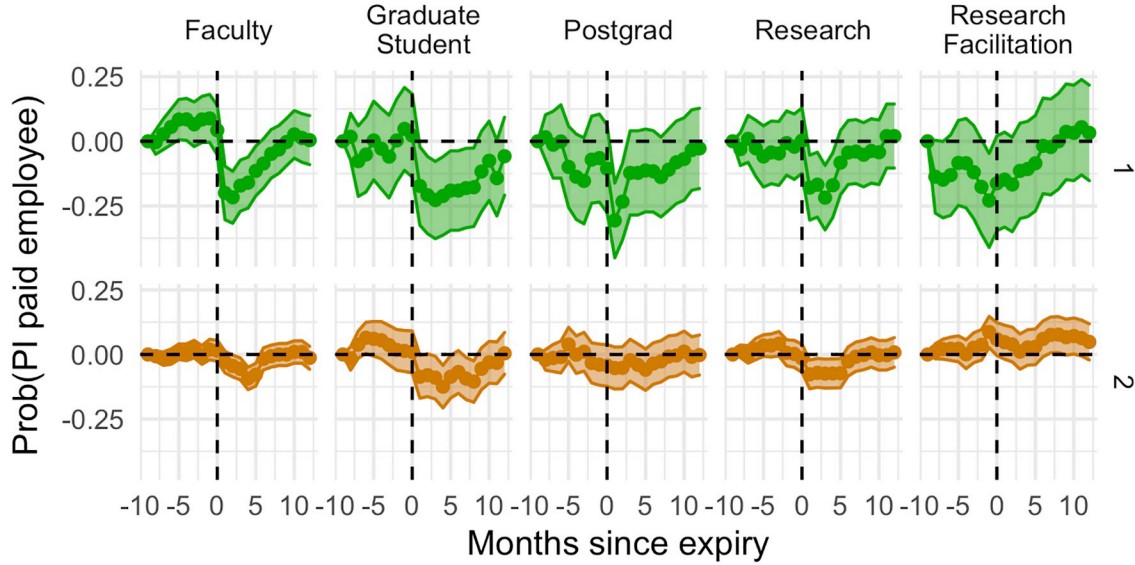

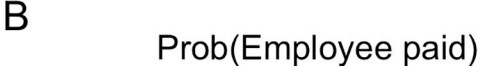

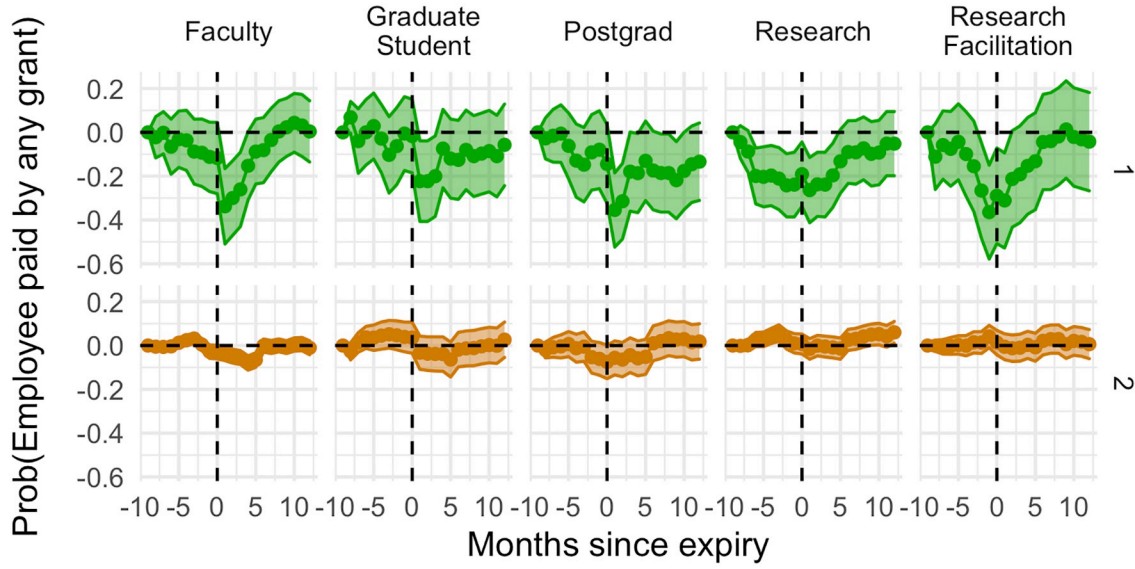

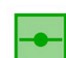

**Fig 4. This figure shows event study coefficients with 95% confidence intervals (clustered by expiring R01 project period) of the difference in probability of being paid for employees on an interrupted project relative to those on an uninterrupted project.** The same event study is estimated on subsamples by occupation and number of R01s. Panel A is for the outcome variable of whether an employee is paid by the PI of the renewed R01. Panel B is for the outcome variable of whether an employee is paid on any grants.

**Table 1. Static difference-in-differences estimates.**

| | No. of Pubs | | Cite-weighted Pubs | |
|---|---|---|---|---|
| | 1 R01 (1a) | 2+ R01 (1b) | 1 R01 (2a) | 2+ R01 (2b) |
| Interrupted-by-Post | 0.002 | -0.016 | 0.013 | -0.024 |
| | (0.016) | (0.015) | (0.029) | (0.025) |
| Num.Obs. | 108900 | 115700 | 108900 | 115700 |
| R2 Adj. | 0.764 | 0.802 | 0.668 | 0.708 |

This table shows the 'static' difference-in-difference estimates and 95% confidence intervals of the difference in publication output if a PI had an interrupted R01. The regression includes treatment-cohort-by-year and PI-R01-renewal fixed effects and uses weights from coarsened exact matching on age, gender, and pre-interruption publications (raw counts and citation-weighted). Dependent variables are raw publication counts and citation-weighted (3-year forward citations) publications, both arcsinh-transformed. Standard errors are clustered by expiring R01 project period. Event study plots are available in the Appendix.

any grant; on the other hand, for those associated with multiple R01s, the probability of being paid decreases less or not at all. Over time, interrupted and uninterrupted employees converge in their probability of being paid. However, for the *Postgraduate*, *Graduate Student*, and *Research* occupations, employees with one R01 remain 13, 6, and 5 percentage points less likely (respectively) to be paid on any grant a year after R01 expiry.

These results raise the question of whether employees in those occupations are paid less or even leave their institution, because it may be harder to find internal sources of funding for them.[28]

## Research output

In my final set of results, I estimate the effect of interruptions on publications at the yearly level. Table 1 shows the estimates from the "static" specification. None of the estimates are statistically different from zero. This result is consistent with the event study figures (Appendix Section), which do not show obvious differential trends or levels in publications before and after an interruption.[29]

One reason for the imprecise estimates may be that interruptions are simply not particularly disruptive in practice. For instance, labs may be able to mitigate the temporary halt in spending by devoting more time to aspects of research that do not immediately need money (e.g. thinking of new ideas, writing).

Another reason for these results may be that universities can sufficiently mitigate the effects of a funding interruption through mechanisms such as bridge funding. In this case, resources have to be diverted to counteract the "true" negative effect of interruptions on research output.

Finally, standard measures of productivity such as publications may be too coarse relative to the true effects of funding interruptions. In addition, variable publication lags may result in the effects of an interruption being "smeared" across several years and therefore hard to detect.

## Conclusion

I study how NIH-funded researchers respond to funding interruptions. Using transaction-level data, I am able to examine these effects at a level of granularity that was previously unavailable. I find evidence that interruptions are disruptive to research. PIs spend less either because of the uncertainty about whether or when they will be funded again, or because they are not able to draw on funds from their next budget, or both. These changes may, in turn, be disruptive to the work and training of employees, who become less likely to be paid on grants.

In ongoing work, we investigate whether this affects a wider range of employment outcomes such as earnings or even having to leave their institution.

These results point to two important policy implications. First, policies to reduce uncertainty can help us to avoid the costs of disruptive events such as funding interruptions. An example would be a multi-year appropriation for funding agencies' budgets. Second, given that some amount of uncertainty is unavoidable, how organizations choose to react to uncertainty is an important policy lever that can be more realistically adjusted. This paper underscores that organizations' risk aversion comes with costs that should factor into decision-making.

## Appendix

This appendix provides additional details on the background and data of the paper, as well as supplementary results not in the main text.

### Additional background

**Scientists' concern about uncertainty.** Uncertainty over funding is a real concern among scientists. DrugMonkey, an anonymous blog run by an NIH-funded researcher, has a post titled "Never Ever Trust a Dec 1 NIH Grant Start Date". The post warns that projects that are due to be funded on December 1—that is, on the first funding cycle of the fiscal year—are rarely funded on time due to delays in Congress passing the budget.

Even well-established researchers report that uncertainty over funding limits their ability to do research. An article in the *San Diego Union Tribune* about the impact of NIH budget uncertainty features a prominent cancer researcher, Dr. David Cheresh, expressing that "(t)he uncertainty that the NIH feels reflects itself in my willingness to hire." Dr. Cheresh is an NIH MERIT awardee with over 70,000 citations, suggesting that even scientists with strong track records are affected by the lack of long-term budget planning.[30]

**Time series of interruptions.** Fig 5 shows the time series of interrupted R01s. For each fiscal year, the graph shows the percentage of R01 projects renewed within the same fiscal year that experienced a greater than 30-day gap between expiry and renewal. There is an increase in the rate of interruptions over time, though with substantial fluctuations around the overall trend. I also highlight the spike in interruptions in Fiscal Year 1996—during there were two federal government shutdowns (including one that lasted three weeks) that were reported to be highly disruptive to the grant making process [3]—as suggestive that the NIH does respond to delays in the federal budgeting process.

**Variation in interruptions across NIH Institutes and Centers.** The NIH is comprised of 27 Institutes and Centers, commonly known as "ICs". Each IC is focused on a particular disease (e.g. National Cancer Institute) or body system (e.g. National Heart Lung Blood Institute). ICs administrate their own budgets and thus may choose to respond to budget uncertainty differently. The National Institute of Allergy and Infectious Diseases (NIAID), for example, describes itself as being "assiduous about issuing awards using funds from the CR (continuing resolution)".[31]

Fig 6 repeats Fig 5, showing the percentage of R01 projects that experienced a greater than 30-day gap, but for two different ICs (NIAID and NCI) rather than for the NIH as a whole. In recent years, NIAID has had a consistently lower proportion of projects experience interruptions than NCI. Even when there was an acute shock to the budgetary process during the 1996 government shutdown, both ICs appear to have responded differently, with NCI having more than 40% of its projects interrupted compared to just over 10% for NIAID.

**Internal funding.** Although internal funding is not observed in UMETRICS, I interviewed a university administrator who works with faculty on grant management to gain a

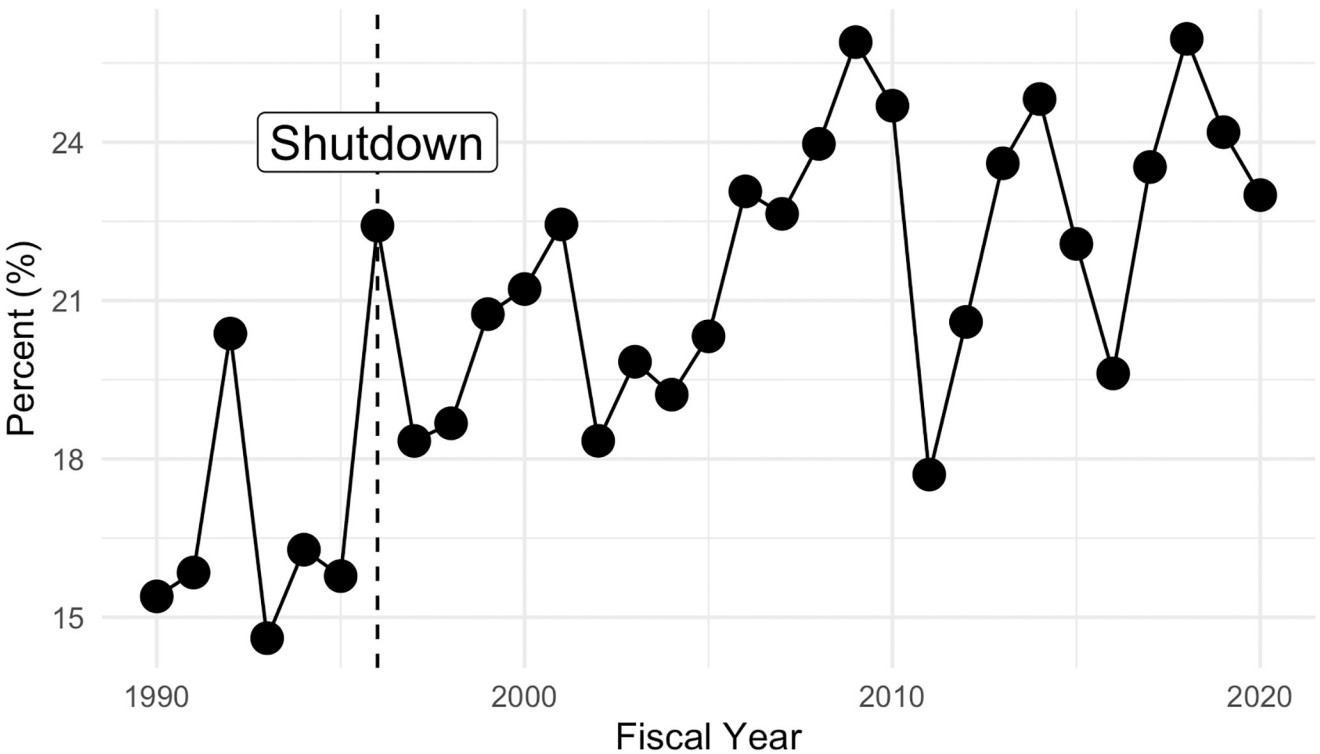

**Fig 5. Proportion of renewed R01s that experienced an interruption by fiscal year.** An interruption is defined as a gap in funding of more than 30 days. Source: NIH ExPorter.

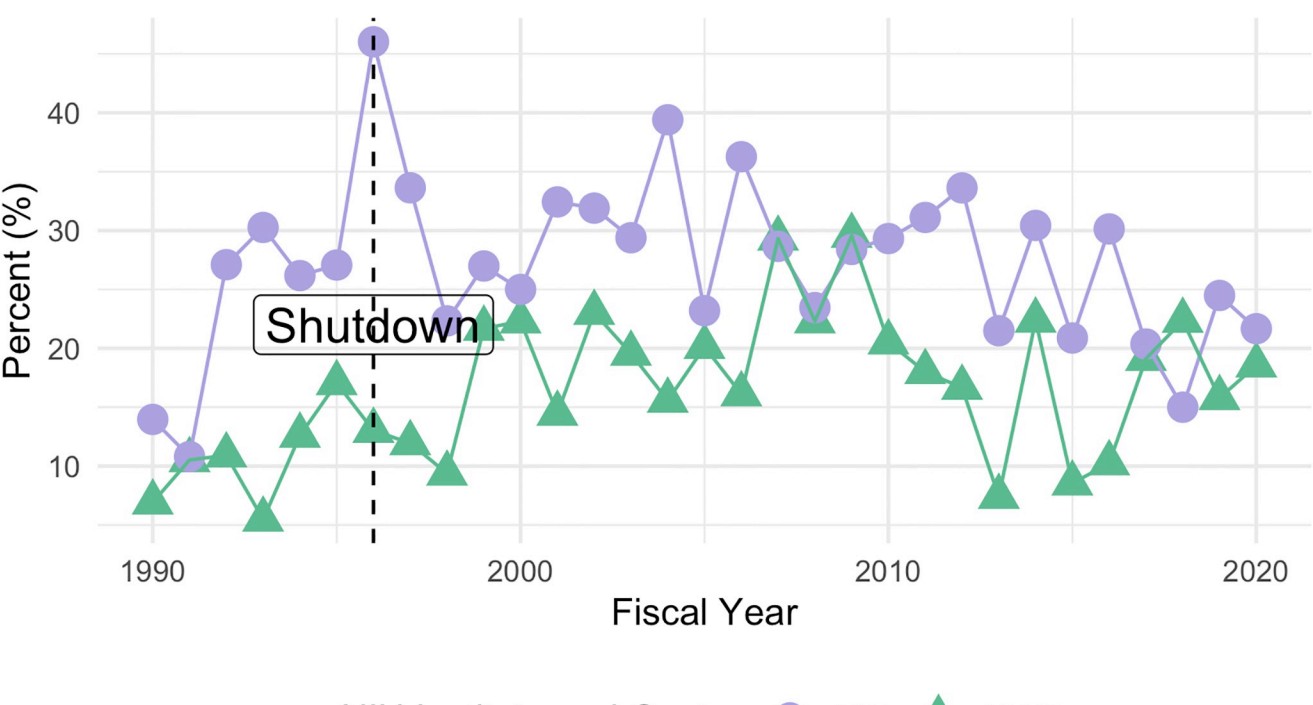

**Fig 6. Variation in interruptions across NIH Institutes and Centers (ICs).** The figure shows the proportion of interrupted projects by fiscal year for the National Cancer Institute (NCI) and National Institute of Allergy and Infectious Diseases (NIAID). Source: NIH ExPorter.

better understanding of how Principal Investigators (PIs) and their institutions react to funding delays. This interview is referenced in the main text and here I provide a fuller description of its content.

The administrator expressed that the university's willingness to provide internal funding varies by individual circumstances such as whether the faculty member is making an effort to find other funding sources or whether they could enter into a collaboration with another faculty member. While they could not explicitly state the extent to which internal funding could make up for the funding gap, they said that it was generally easier to get internal funding for non-personnel and "essential" expenditures such as live animals. Conversely, they said it was harder to get internal funding for personnel and a PI facing funding difficulties might therefore be advised to downsize their lab. In this situation, graduate students may find a new mentor and/or teach instead, while postdocs would have to find a new job.

## Data

The paper uses three main data sources.

1. NIH ExPorter

2. UMETRICS (2019 release)

3. Author-ity

### NIH ExPorter

NIH ExPorter is publicly available data from the NIH that can be found at https://exporter. nih.gov/. ExPorter provides the following types of data that can be linked to each other: Projects, Project Abstracts, Publications citing support from projects, Patents citing support from projects, Clinical Studies citing support from projects.

**Defining project periods.** NIH projects are assigned a **core project number** that is used over multiple **project periods**. The funds for a project period are allocated from the NIH to the project over multiple **budget periods**.[32] Each budget period is recorded as a row in the ExPorter *Projects* data. However, ExPorter does not provide identifiers for project periods. The rest of this section explains how I construct these identifiers.

At the end of each project period, they can apply to renew funding for that project for a new project period. Thus, a project can be last for multiple project periods.

Although project periods last 4–5 years, the funds for a project are technically released over multiple *budget periods*. Each budget period is typically a year in length. ExPorter reflects this by having a new row for each time a project funds are allocated to a project. For example, project number *R01GM049850*, led by PI Jeffrey A. Simon, was funded from FY 1996 to FY 2017, except for FY 2013. Table 2 below shows the first two project periods that it was funded.

The NIH makes data on awarded grants publicly available through its ExPorter database. While projects can be identified through their R01 *core project numbers*, there is no explicit identifier for project periods. I describe below how I define project periods using ExPorter variables and data structure.

The key to defining project periods is using the *Application Type* variable. This is a one-digit code that describes the type of "application" funded (see Table 3 for a full list of application types). For our purposes, the application type allows us to distinguish between what the NIH calls "competing" and "noncompeting" awards. "Competing" funds are provided as a result of having gone through a competitive process against other grant application. "Noncompeting" funds are provided as part of an already awarded project period. For the typical project,

**Table 2. Example of NIH ExPorter data before aggregation into project periods.**

| PI Name | Core Project Num | Fiscal Year | Application Type | Comment |
|---|---|---|---|---|
| Simon, Jeffrey A | R01GM049850 | 1996 | 1 | New |
| Simon, Jeffrey A | R01GM049850 | 1997 | 5 | Continuation |
| Simon, Jeffrey A | R01GM049850 | 1998 | 5 | Continuation |
| Simon, Jeffrey A | R01GM049850 | 1999 | 5 | Continuation |
| Simon, Jeffrey A | R01GM049850 | 2000 | 2 | Renewed |
| Simon, Jeffrey A | R01GM049850 | 2001 | 5 | Continuation |
| Simon, Jeffrey A | R01GM049850 | 2002 | 5 | Continuation |
| Simon, Jeffrey A | R01GM049850 | 2003 | 5 | Continuation |

**Table 3. Full list of application types for NIH grants.** Detailed definitions available at https://grants.nih.gov/grants/how-to-apply-application-guide/prepare-to-apply-and-register/type-of-applications.htm.

| Type | Stage |
|---|---|
| 1 | New |
| 2 | Renewal |
| 3 | Competing Revision |
| 4 | Extension |
| 5 | Noncompeting Continuation |
| 6 | Change of Organization Status (Successor-in-Interest) |
| 7 | Change of Grantee or Training Institution |
| 8 | Change of Institute or Center |
| 9 | Change of Institute or Center |

funds disbursed in the first year (i.e. just after the application process) are competing and funds awarded in subsequent years are noncompeting.

I identify R01 project periods as follows:

1. Identify all budget periods with an application type of 1, 2, or 9. These are taken to be the beginning a project period.

2. Assign a set of budget periods to the same project period if they begin in-between the beginnings of two project periods that belong to the same project.

3. Take the beginning of the budget period to be the start of the first budget period

4. Take the end of the budget period to be the end of the budget period that ends the latest. If the budget period ends after the beginning of the next project period, assign the end of the budget period to be one day before the next project period starts.

**Monthly panel in UMETRICS.**

1. Identify R01s renewed within the same fiscal year

2. Identify all PIs associated with each renewed R01

3. For period of interest (for example, 12 months before and after R01 expiry), create a monthly panel for each PI-R01 renewal

4. Restrict panel months that (a) are covered by NIH ExPorter and (b) are covered by in all 3 UMETRICS datasets (award, vendor, subaward)

5. Restrict to expiring R01 project periods that lasted for 6 years or less

6. Restrict to PI IDs that appear in the expiring project period *and* the renewed project period

**Yearly panel for publication outcomes.** I use a similar "stacking" procedure as described above to construct a PI-R01-renewal by year panel that starts 4 years before an interruption and ends 5 years after, restricted to all interruptions that took place from 1989 to 2006. The earliest possible year in the panel is 1985, the earliest year in ExPorter. The latest possible year in the panel is 2011, so the latest possible citation for a 3-year forward citation window is from 2013, which is the final year indexed in the version of Web of Science that I use. For each treatment cohort (indexed by interruption year), I exclude units that were interrupted less than 5 years before the beginning of the cohort to reduce the possibility that previous interruptions might affect the estimates. Finally, if a PI has multiple R01s renewed within the same year, I assign the PI's interruption status based on the R01 with the longest gap between expiry and renewal.

**NIH coverage of overall grant portfolio.** My measure of PI spending is limited to spending through NIH grants. This restriction ensures a high degree of accuracy in linking NIH PIs to transactions. As discussed in the main text, the results on whether interrupted employees continue to be paid on *any grant* are consistent with the overall set of results and thus do not give us a reason to think that there is a substantial pool of non-NIH grants being used to offset the effects of interruptions.

Other research also suggest that focusing on NIH funding provides substantial coverage of researcher funding. [21] estimate that about 70% of research groups (as defined by a community detection algorithm) in the UMETRICS data rely on federal funding for 90% of their funding. [22] estimate that in 2002, the NIH was by far the largest funder of biomedical research, funding over $20 billion in research compared to $1.2 billion by the Department of Defense. More recent data from the Survey of Federal Funds for Research and Development show that in the life sciences, the NIH has provided about 80% of federal funding for research (basic and applied research combined) in colleges and universities since 2003.[33]

**Negative transaction amounts in UMETRICS.** Some transactions in UMETRICS are negative amounts. These can appear for a number of reasons including returns, discounts, reversing a purchase that was wrongly assigned, or money that was unused and refunded. In general, it is not possible to separately identify these reasons. If the negative amount is related to a purchase it is also not possible to identify that purchase (e.g. in the case of discounts or returns). Thus, I treat negative amounts as occurring at the transaction date when summing up transaction amounts to the PI-month level. If expenditure in a PI-month remains negative after summing up, I assign a value of zero. In the final sample, I also exclude PIs that have an unusually high amount of negative expenditure relative to the rest of the sample. Specifically, over the 24-month period covered by the panel, I sum up across months where total expenditure was negative and then across all months where total expenditure was positive. I exclude a PI if the absolute value of total negative expenditure was greater than or equal to the absolute value of total positive expenditure. Fig 7 shows the distribution of the ratio of total negative to total positive expenditure.

## UMETRICS

I use the 2019 release of the UMETRICS data set, which is housed at IRIS (Institute for Research on Innovation & Science). In this appendix I describe the most relevant components

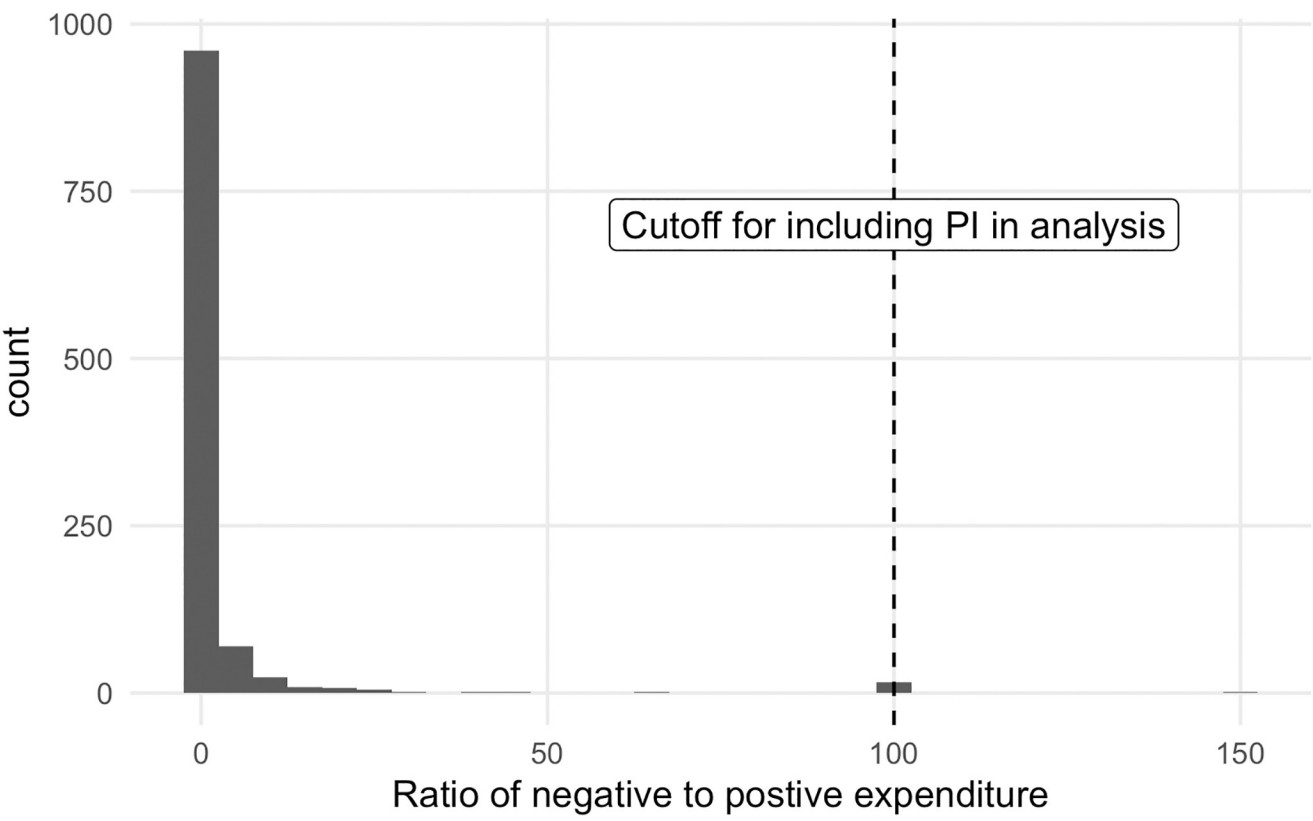

**Fig 7. This is a histogram of the ratio of total negative to total positive expenditure amounts for a PI, as described in the section on negative transaction amounts in UMETRICS.** The ratio is given a value of zero if total positive expenditure was zero and total negative expenditure is also zero.

of the dataset to this paper. A summary documentation of the data is publicly available at this link. The UMETRICS Core Collection consists of administrative data from universities "drawn directly from sponsored projects, procurement, and human resources data systems". The Core Collection consists of four datasets: award, vendor, subaward, employee. "Award" data record the total expenditure from an award in a given transaction period, while the "vendor" and "subaward" data record payments to a vendor and subaward in a given transaction period respectively. "Employee" data record when an employee is paid by an award, but do not contain information on wages. In the analysis, payments to labor are backed out as the remainder after subtracting vendor and subaward payments from total expenditure (i.e. *Labor = Total − Vendor − Subaward*).

In addition to the Core Collection, there is also an Auxiliary Collection and Linkage Collection that consist of data linking the Core Collection to information such as institution characteristics or external grant data such as NIH ExPorter.

The 2019 UMETRICS release consists of data from 31 Universities. I restrict the sample to projects in institutions where transaction periods are at the monthly level. For employee data, pay periods that last longer than a month are converted to monthly, assuming employed each month contained in the quarter.

**Employee occupational classifications.** Figs 8 and 9 (from the UMETRICS 2019 data manual) describe in detail the occupational classification categories used in UMETRICS 2019, along with examples of job titles that fall under each category.

## TIER 1 DESCRIPTIONS

| CATEGORY | Description | Example Job Titles |
|---|---|---|
| FACULTY | Advanced academic employees who are directly involved in scientific research, clinical care, and/or scientific instruction. Faculty have the highest degree in their area of study and tenure or a tenure-track/"permanent" position at their university (e.g., in the case of medical school faculty) | Academic Administrators, if faculty (e.g., Dean, Provost, Center Director) Adjunct Professor Clinical Faculty Professor (Assistant, Associate, Full and Chaired) Research Faculty Visiting Professor |
| POST GRADUATE RESEARCH | Individuals holding terminal degrees (PhD, MD) who are in temporary training status | Clinical Fellow Medical Resident/Intern/Fellow Postdoctoral Fellow/Researcher Research Associate |
| GRADUATE STUDENT | Students earning advanced degrees | Graduate Student Medical/Dental/Nursing Student Research Assistant |
| UNDERGRADUATE | Students earning baccalaureate/other degrees including full-time, part-time, summer research assistants, and work-study. Includes high school students who would likely be acting in a similar capacity | Intern Nursing Student (BA programs) Student Worker Undergraduate Student |
| STAFF | Individuals in non-faculty roles who are clinical staff, research and research support staff, instructional staff, and technical support staff | (See Table 16 below) |
| OTHER | Positions that support general university functions such as undergraduate education and student activities. Employees whose titles cannot be attributed to the scientific research enterprise | |

**Fig 8. This is one of two screenshots of tables from the UMETRICS 2019 manual describing the employee occupation categories.**

## TIER 2 DESCRIPTIONS

| CATEGORY | Description | Example Job Titles |
|---|---|---|
| CLINICAL | All non-faculty health care professionals (clinicians) | Clinical Psychologist<br>Dental Hygienist<br>Dietician or Nutritionist<br>Genetics Counselor<br>Nurse<br>Physical Therapist<br>Social Worker |
| RESEARCH | Non-faculty scientists, engineers, analysts and technicians. Research staff are directly involved in conducting research, usually have advanced degrees, and are skilled and specialized in some area of science, technology, equipment or research but are not faculty members | Engineer<br>Lab Manager<br>Medical Technician<br>Regulatory Officer<br>Research Analyst<br>Research Associate, Specialist, or Professional<br>Staff Scientist<br>Statistician<br>Technician |
| RESEARCH FACILITATION | Administrative employees who are not specifically employed for scientific research purposes but perform job tasks that support the research enterprise. Individuals who serve as managers/coordinators/facilitators for laboratory studies/clinical trials/large facilities/research programs. Employees who direct and influence scientific research activity from the level of the laboratory up to the level of the university/research center | Administrative Staff<br>Associate Center Director<br>Associate Dean or Provost<br>Communications Specialist<br>Grants Manager<br>Interviewer<br>Lab Coordinator (not Lab Manager)<br>Laboratory Aide<br>Managing Director<br>Operations Manager<br>Project Manager<br>Study Coordinator |
| TECHNICAL SUPPORT | Technical employees who are not specifically employed for scientific research purposes but perform job tasks that support the research enterprise | Data Entry/Data Analyst<br>Information Technology Manager<br>Network Support Specialist<br>Programmer<br>Software Engineer |
| INSTRUCTIONAL | Staff members who are instructional or academic specialists | Academic Specialist<br>Instructor<br>Lecturer |
| OTHER STAFF | All other research staff that do not clearly fall into another category | |

**Fig 9. This is one of two screenshots of tables from the UMETRICS 2019 manual describing the employee occupation categories.**

### Inverse hyperbolic sine

Unless otherwise stated or the outcome is a binary variable, I apply an inverse hyperbolic sine (also *asinh* or *arcsinh*) transformation to the outcome variable for all regressions, which approximates a natural logarithm and is defined at zero. The approximation is worse at smaller values [23]. For "large" outcomes i.e. spending amounts, I convert estimates to percentage changes using the standard $exp(\hat{\beta}) - 1$ for log transformations. When the outcome variable is "small" (e.g. for counts of employees in a lab), I use the mean of the *arcsinh*-transformed outcome variable for interrupted PIs ($asinh(y_0)$) to back out the percentage change as follows:

$$\frac{y_1}{y_0} - 1 = \frac{sinh(asinh(y_0) + \hat{\beta})}{sinh(asinh(y_0))} - 1$$

## Additional descriptive statistics and results

### Distribution of renewal gap if not interrupted

Fig 10 is a histogram of the time between expiry and renewal for R01s from the UMETRICS analysis sample that were *not* interrupted, i.e., renewed within 30 days. The distribution is

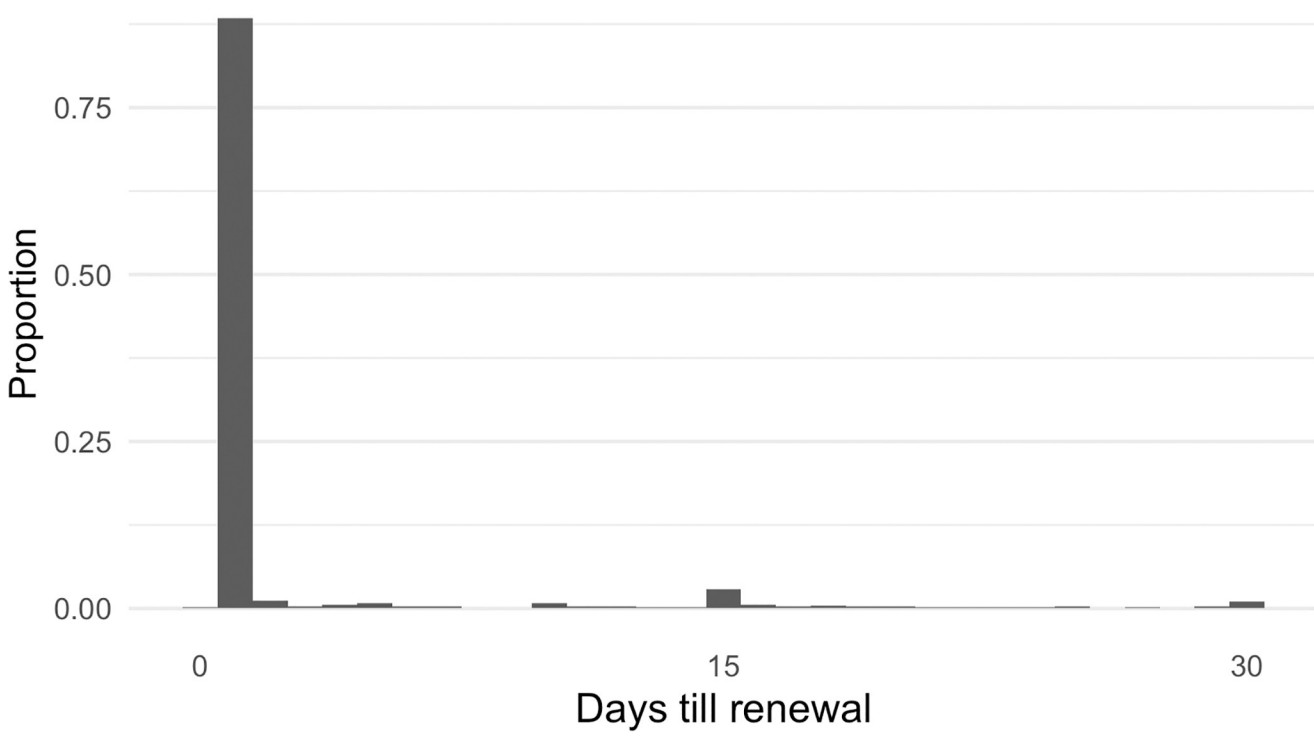

**Fig 10. This figure shows the distribution of the time between expiry and renewal for R01 project periods that were not interrupted, i.e., renewed within 30 days.** Each unit of observation is an R01 project period. The figure is a histogram with 1-day bins.

# Proportion of R01 renewals by year

A

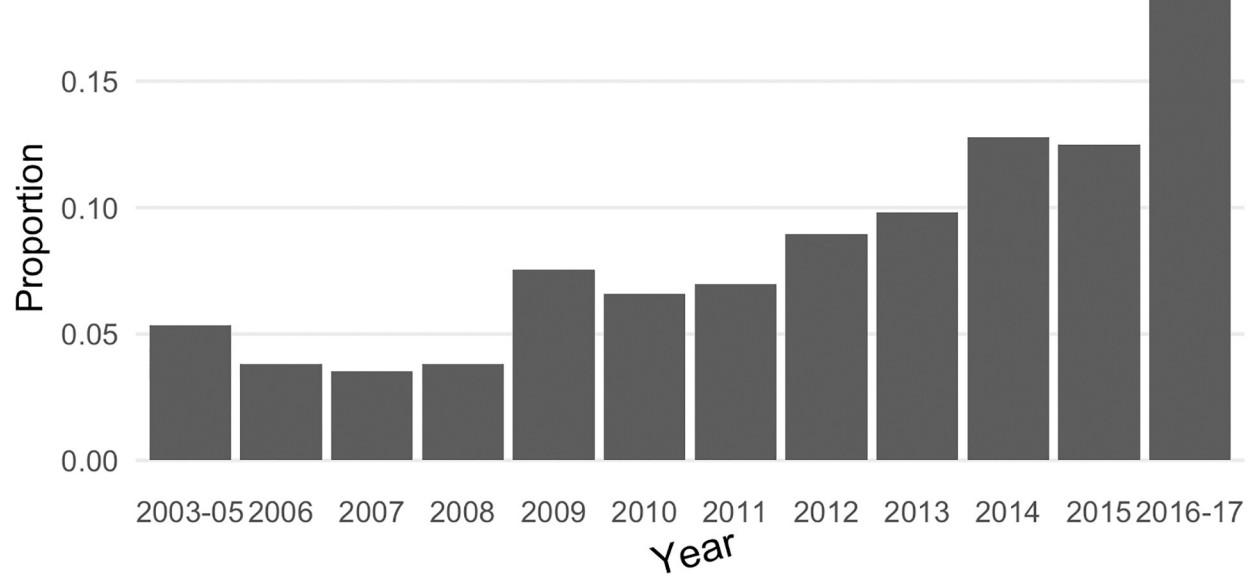

B

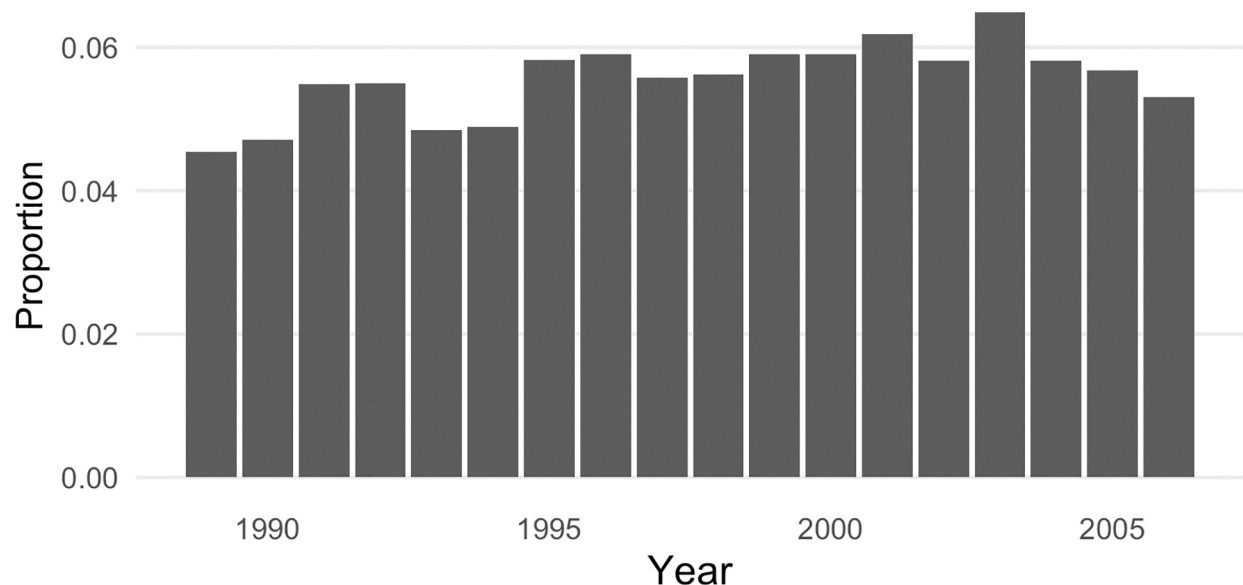

**Fig 11. This figure shows the timing of R01 expiry for each PI-R01 combination in the analysis samples.** Figure A shows the proportion of PI-R01s from the UMETRICS sample where expiry occurred in a given year, while Figure B shows the same for the ExPORTER sample.

concentrated at 1 day because funding tends to start on the first day of the month. There is another small spike at 15 days, which similarly is because the fifteenth of the month is the next most common day to start funding.

### Timing of R01 expiry

Fig 11 shows the timing of R01 expiry in each of the samples used in the paper. Fig 11A shows the proportion of R01 expiries occuring in each year for the UMETRICS sample, while Fig 11B shows the same for the ExPORTER sample (where the outcome variable was publications).

### Spending

**Event study without matching.** Fig 12 repeats the main event study estimation without matching and the results are similar.

**Matching on NIH IC and university.** I repeat the event study estimation, this time matching NIH Institute and Center (IC) and university. This procedure substantially reduces the sample size but the results remain similar (Fig 13).

**Spending distribution.** Fig 14 shows the average *arcsinh*-transformed spending per month for interrupted and interrupted projects. For labs with one R01, uninterrupted labs decrease spending in the months before grant expiry, which then undergoes a gradual increase

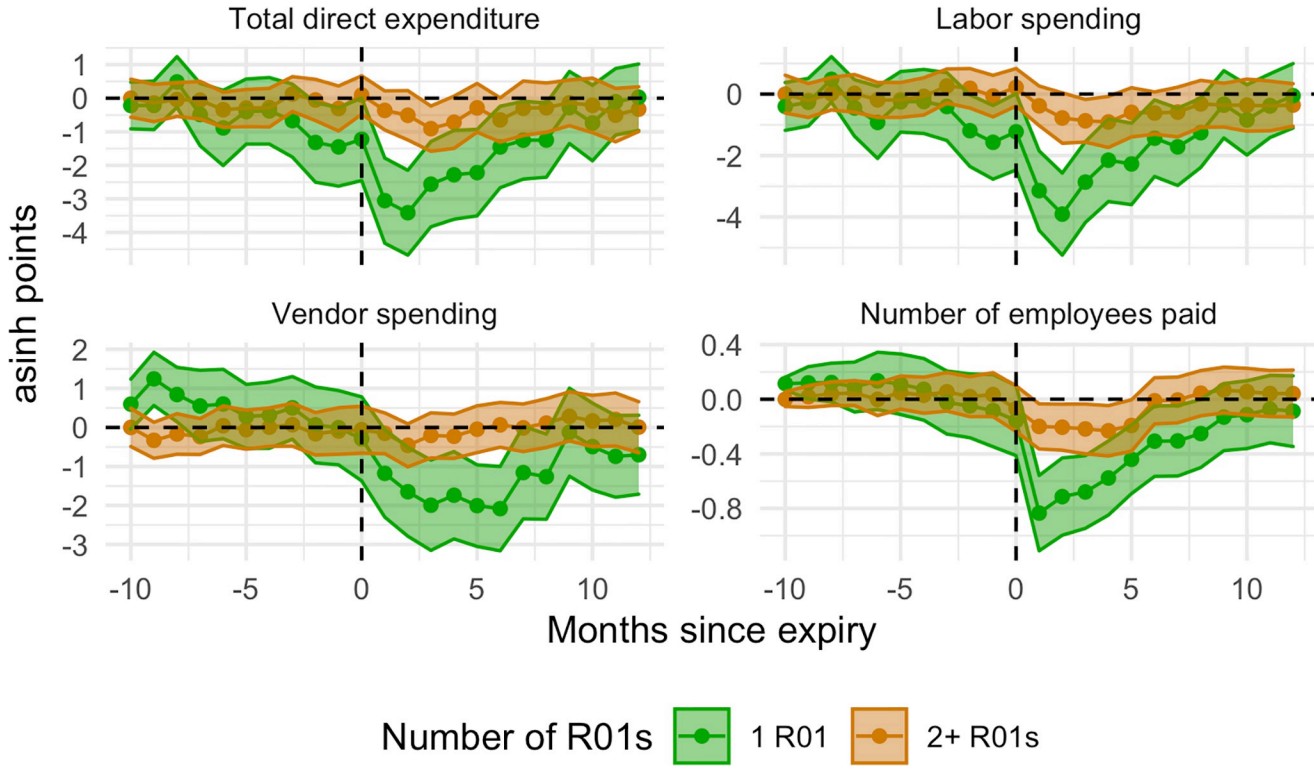

**Fig 12. This figure shows event-study estimates (with 95% confidence intervals) of the difference in spending between PIs of interrupted and uninterrupted R01s.** Each panel shows the estimates for a different outcome variable: total expenditure by PI, total vendor expenditure by PI, total labor expenditure by PI, and total number of employees paid by PI. Month 0 is the month that the expiring R01 expires. Month -11 is the excluded category for the regression. Regressions are run separately on subsamples of PIs that have one R01 grant (green) or multiple R01s (brown), including R01-equivalents and P01 grants. Standard errors are clustered by expiring R01 project period.

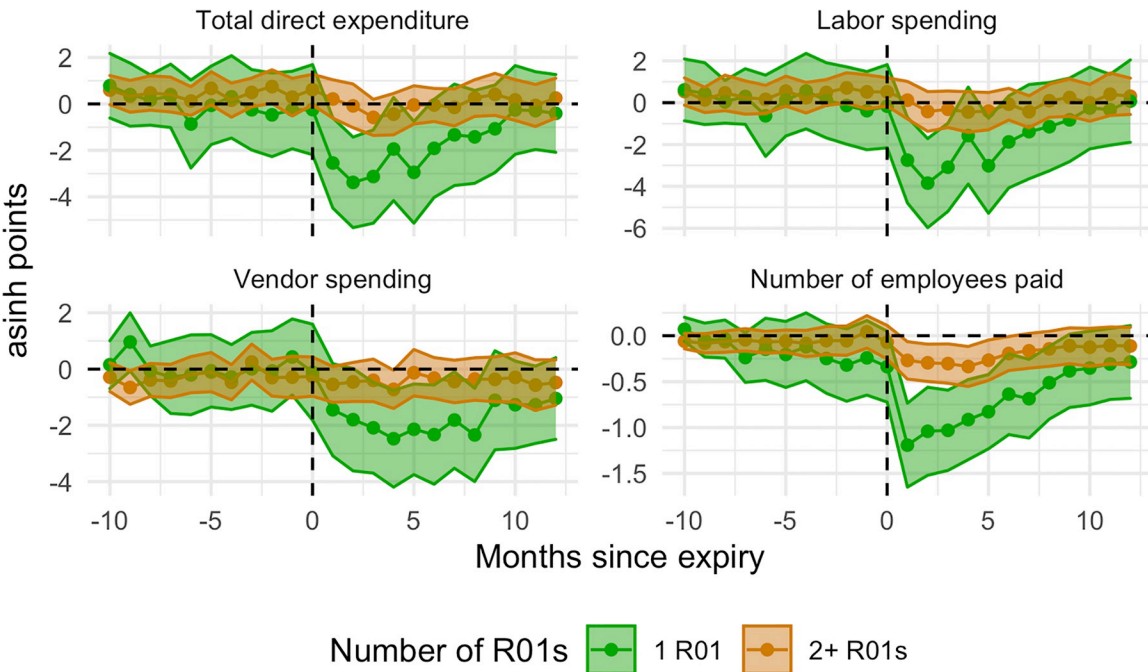

**Fig 13. This figure shows event-study estimates (with 95% confidence intervals) of the difference in spending between PIs of interrupted and uninterrupted R01s.** Treated and control groups are matched on length of the expiring R01 project period, NIH IC, and university, and the regression is weighted using the matching weights. Each panel shows the estimates for a different outcome variable: total expenditure by PI, total vendor expenditure by PI, total labor expenditure by PI, and total number of employees paid by PI. Month 0 is the month that the focal R01 expires. Month -11 is the excluded category for the regression. Regressions are run separately on subsamples of PIs that have one R01 grant (green) or multiple R01s (brown), including R01-equivalents and P01 grants. Standard errors are clustered at the expiring R01 level.

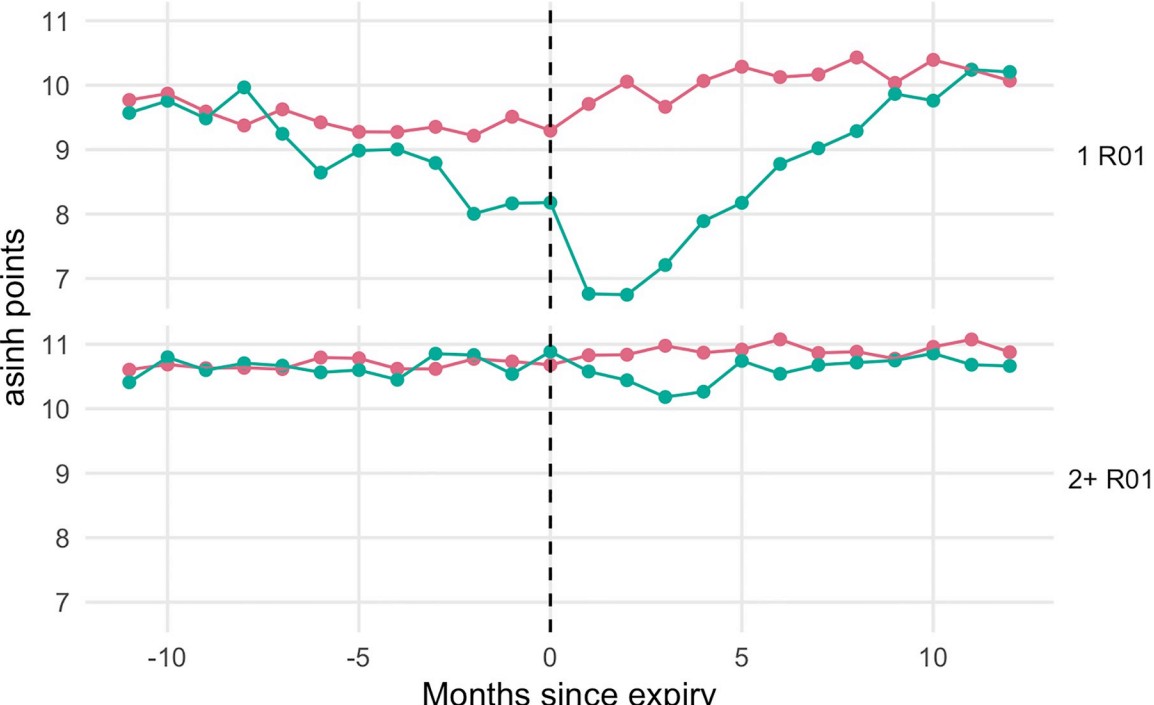

**Fig 14. Average total direct expenditures (arcsinh transformed) per month for interrupted and uninterrupted projects, separately calculated for Principal Investigators with one R01 and those with at least two R01s.**

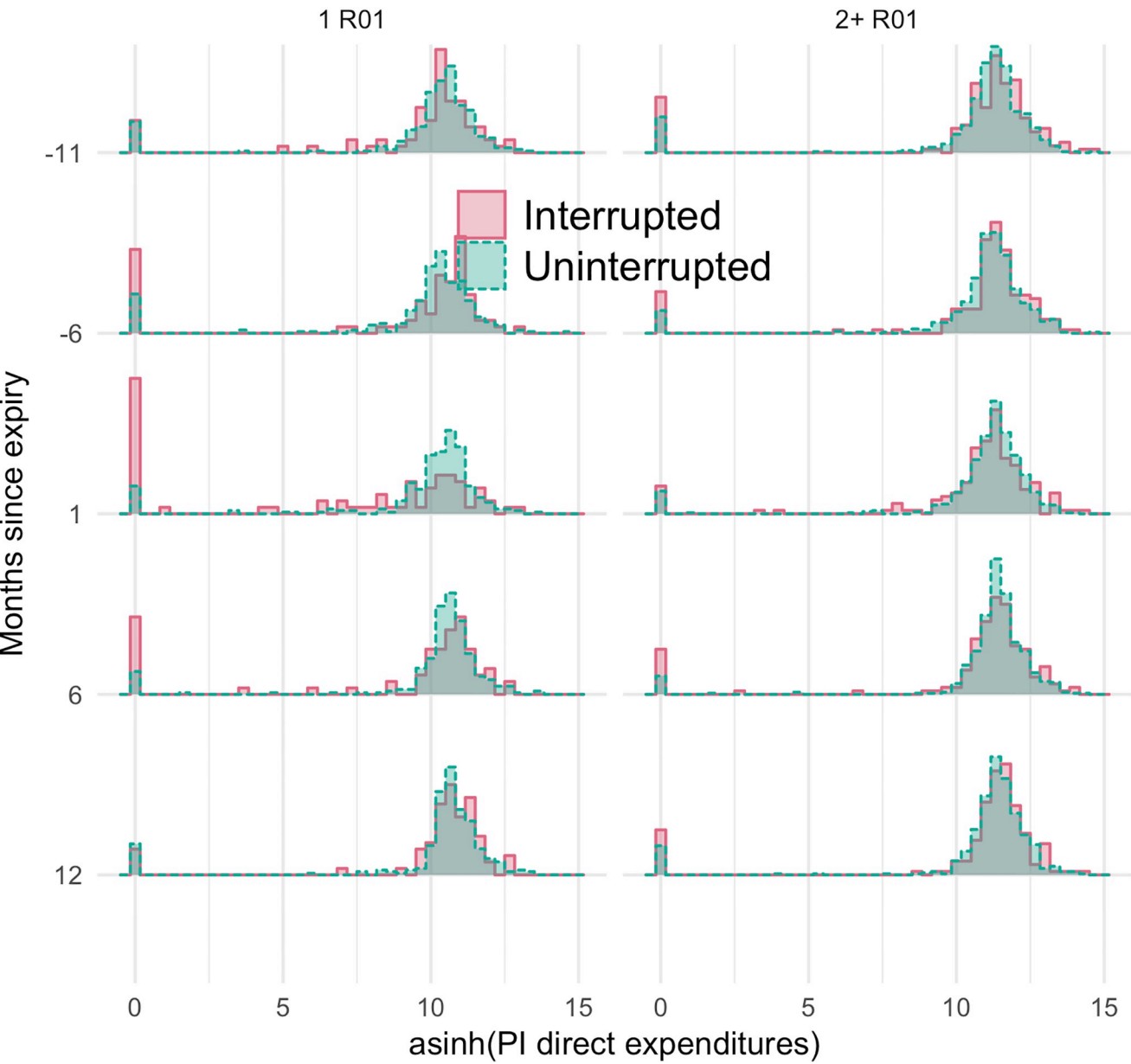

**Fig 15. Histogram of total direct expenditures for each month relative to R01 expiry.** Unit of observation is a PI-R01 period.

with the beginning of the new grant period. Interrupted labs also decrease spending before expiry but the decrease is much more pronounced. In addition, the drop in spending continues into the first month after expiry before recovering.

**Distribution of spending.** Fig 15 shows how the entire spending distribution changes over time. For clarity, I only show select months. The decrease in spending is driven by a "spreading" of the distribution, rather than a shifting. This results in a mass of PIs at zero, but

there also remain a substantial portion of interrupted PIs that continue to spend similar amounts to uninterrupted PIs.

**Length of interruption.** I repeat the event-study analysis, allowing the length of the interruption to vary by estimating separate coefficients for interruptions that lasted 31 to 90 days and interruptions that were more than 90 days.

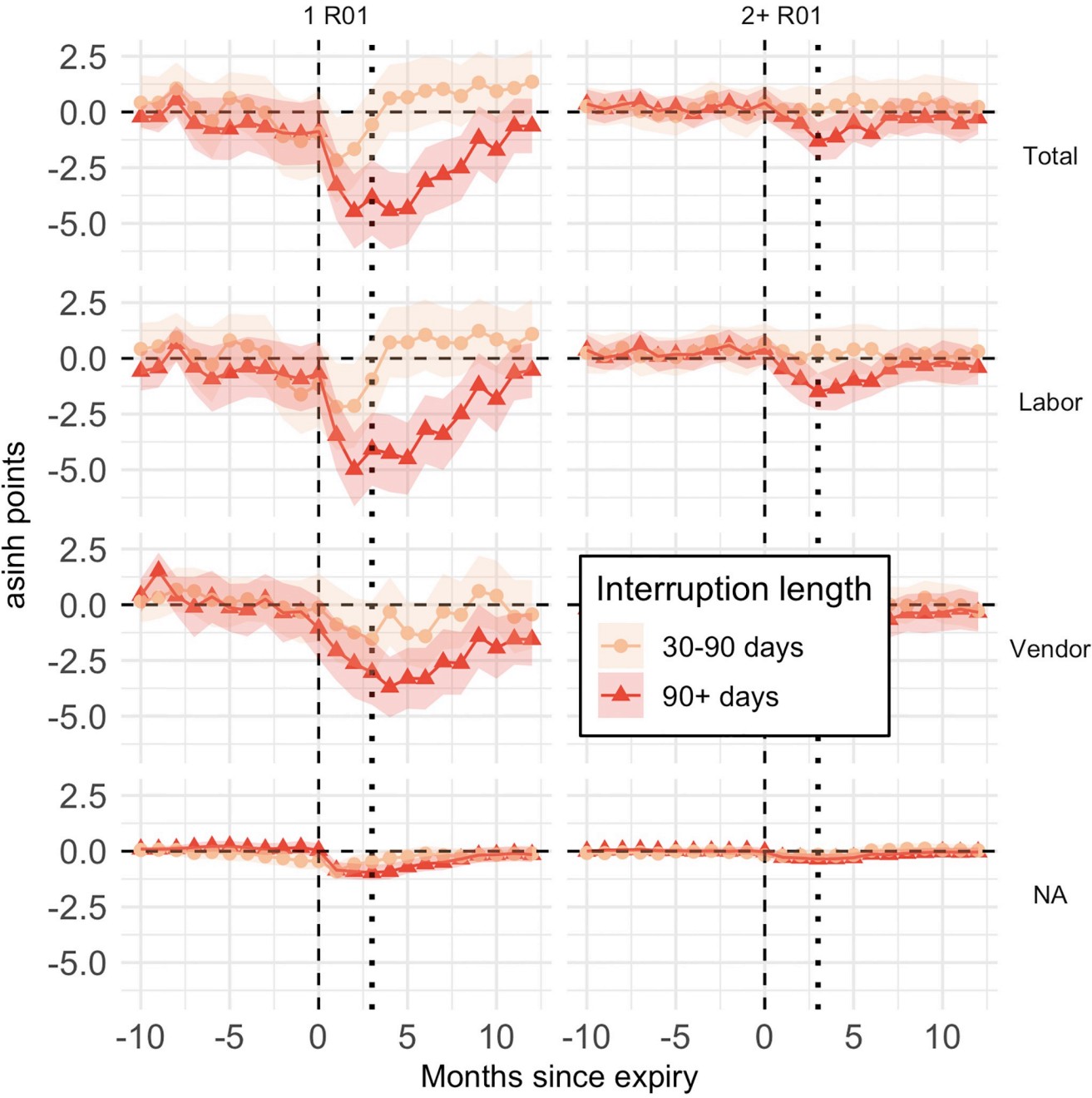

**Fig 16. This graph shows event-study estimates from a balanced panel of R01-PIs 12 months before and after the focal R01's expiry month, covering a period of 24 months.** Separate event study coefficients are estimated for interruptions that were 31 to 90 days and interruptions that were more than 90 days. The regressions include R01-PI fixed effects and relative-to-expiry month fixed effects. Month 0 is the month that the project's budget expires. These regressions are run separately on subsamples of PIs that have one R01 grant (left) or multiple R01s (right), including R01-equivalents and P01 grants. Month -11 is the excluded category for the regression. 95% confidence intervals are clustered by expiring R01 project period.

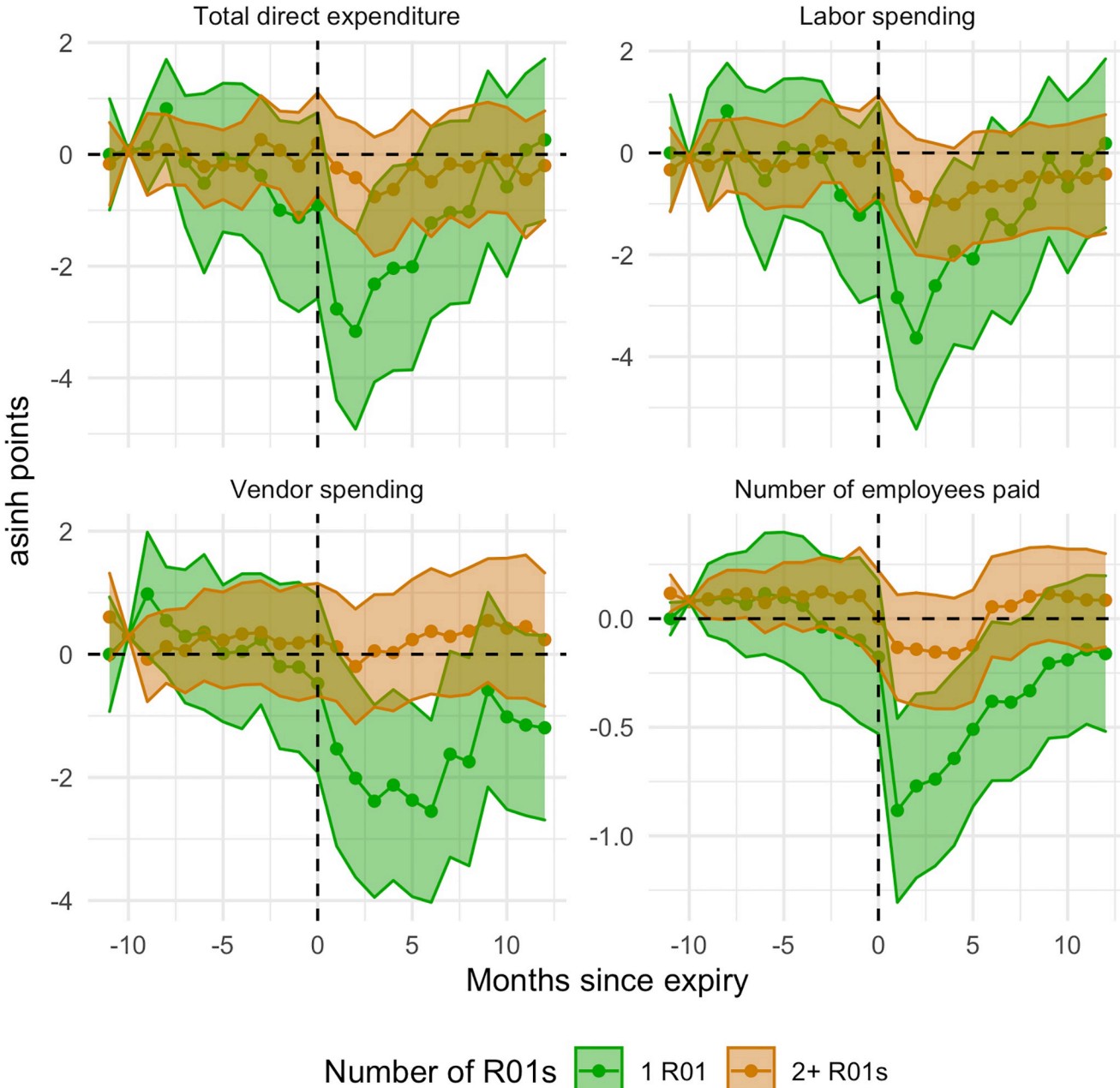

**Fig 17. This figure shows event study estimates (with 95% confidence intervals) of the difference in spending between PIs of interrupted and uninterrupted R01s.** Estimation is done using the Callaway-Sant'anna (2020) doubly robust estimator, including the length of the expiring R01 project period as a covariate. Each panel shows the estimates for a different outcome variable: total expenditure by PI, total vendor expenditure by PI, total labor expenditure by PI, and total number of employees paid by PI. Month 0 is the month that the focal R01 expires. Month -10 is the excluded category for the regression. For comparison with the main results, the point estimates and confidence intervals have been adjusted so that the point estimate for Month -11 is 0. Regressions are run separately on subsamples of PIs that have one R01 grant (green) or multiple R01s (brown), including R01-equivalents and P01 grants. Standard errors are clustered by expiring R01 project period.

## PI Response to Interruption

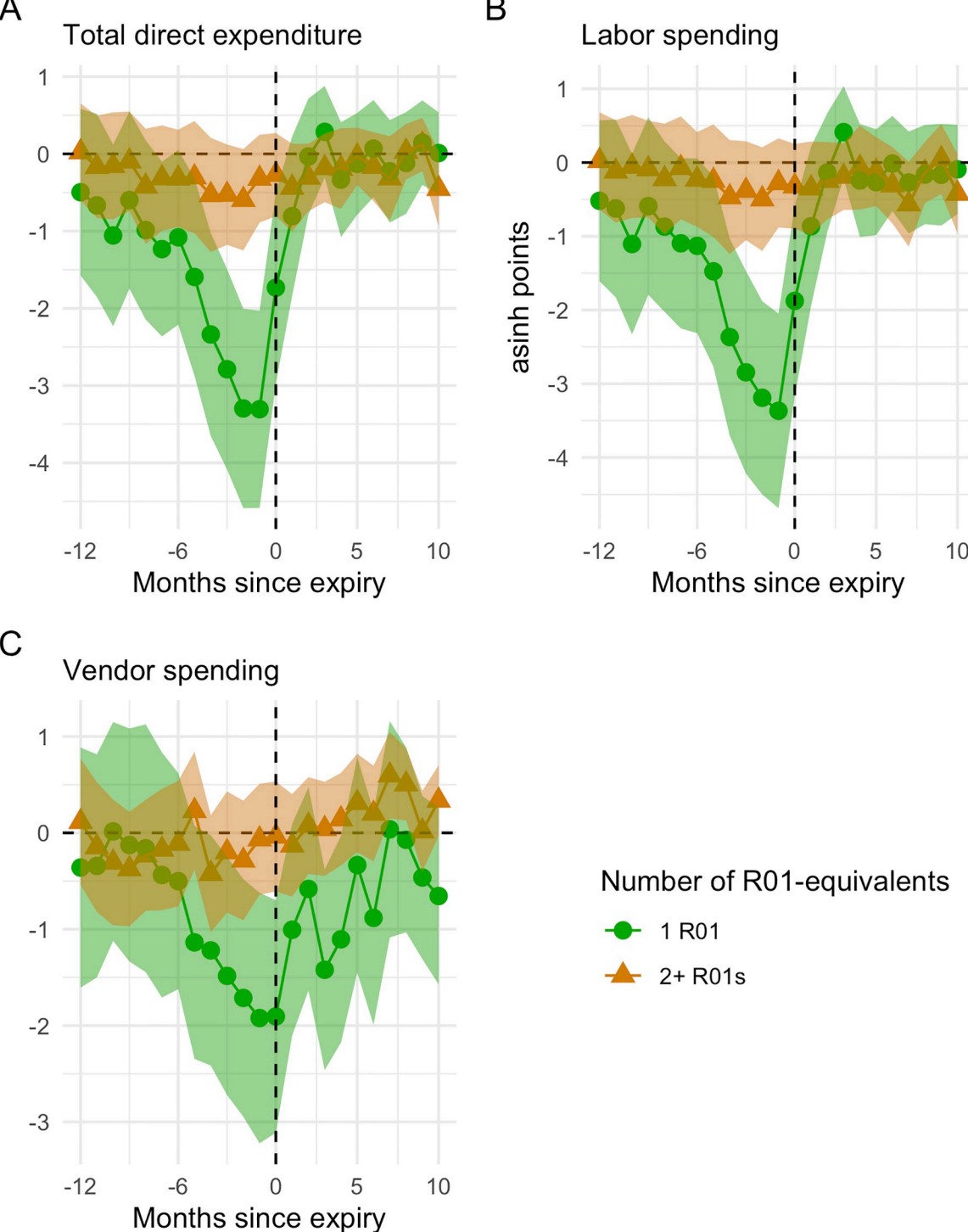

**Fig 18. This figure shows event-study estimates (with 95% confidence intervals) of the difference in spending between PIs of interrupted and uninterrupted R01s.** Each panel shows the estimates for a different outcome variable: total expenditure by PI (A), total vendor expenditure by PI (B), total labor expenditure by PI (C), and total number of employees paid by PI (D). Month 0 is the month that the focal R01 expires. Month 11 is the excluded category for the regression. Regressions are run separately on subsamples of PIs that have one R01 grant (green) or multiple R01s (brown), including R01-equivalents and P01 grants. Standard errors are clustered by expiring R01 project period.

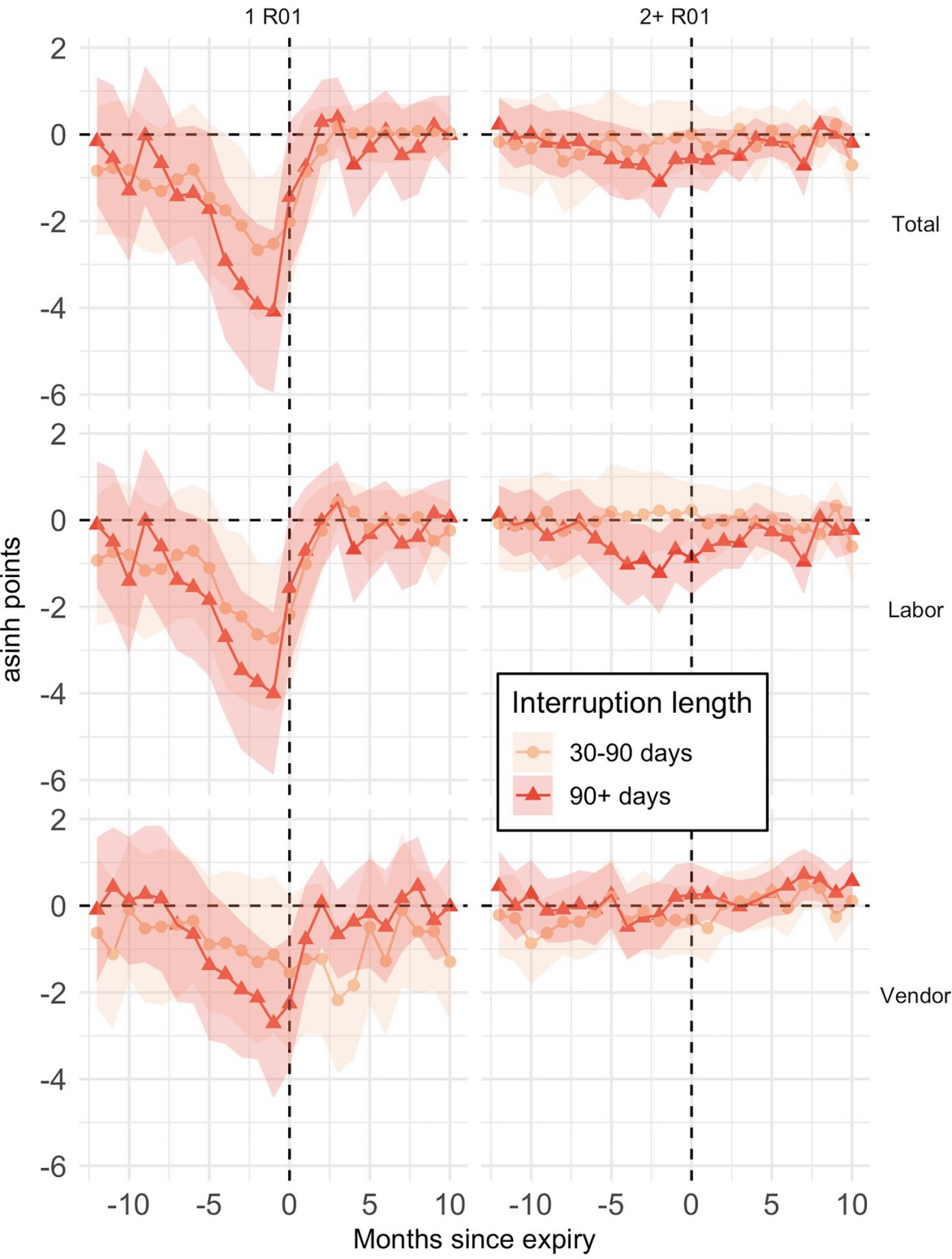

**Fig 19. This graph shows event-study estimates from a balanced panel of R01-PIs 12 months before and after the focal R01's renewal month, covering a period of 24 months.** Separate event study coefficients are estimated for interruptions that were 31 to 90 days and interruptions that were more than 90 days. The regressions include R01-PI fixed effects and relative-to-expiry month fixed effects. Month 0 is the month that the project's budget was renewed. These regressions are run separately on subsamples of PIs that have one R01 grant (left) or multiple R01s (right), including R01-equivalents and P01 grants. Month 11 is the excluded category for the regression. 95% confidence intervals are clustered by expiring R01 project period.

I index PIs as $L$, R01s as $R$, and the year-month as $t$. $t_{expiry}$ is the year-month that the R01 grant $R$ expires. $e$ is the number of months before expiry i.e. $e = 0$ when $R$ expires amd $e < 0$ before the grant expires. I restrict the sample to the one year before and after the R01 $R$ expires, i.e. $e$ starts at month $-11$ and ends at 12. $e = -11$ is excluded from the specification. The specification is:

$$y_{LRt} = \sum_{e=-10}^{12} \beta_{m1} \mathbf{1}(e = t - t_{expiry}) \mathbf{1}(Interrupted \in (30, 90]) +$$

$$\sum_{e=-10}^{12} \beta_{m2} \mathbf{1}(e = t - t_{expiry}) \mathbf{1}(Interrupted \in (90, \infty)) +$$

$$\delta_{LR} + \gamma_e + \epsilon_{LRt}$$

Fig 16 displays the coefficients. For labs with only one R01, longer interruptions lead to greater drop in spending and a longer recovery. This accords with the intuition that a longer interruption would mean a longer time without access to funding. However, even in Month 3, spending does not recover completely for interruptions lasting between 31 to 90 days, indicating that even when funding becomes available, labs may need time to scale up their work again. This is confirmed in the next subsection with an analysis centered around the date of R01 renewal instead of R01 expiry.

In addition, allowing the length of interruptions to vary reveals that even for labs with multiple R01s, spending is affected by longer interruptions. While the difference is still smaller than for labs with one R01, it is still substantial. For interruptions of more than 90 days, spending decreases by 73% at the lowest month.

**Callaway-Sant'anna estimator.** I repeat the main event study estimation (i.e. with research inputs as outcome variables) with the [13] doubly robust estimator, including the length of the expiry R01 project period as a control. The results are similar (Fig 17).

Note that the main event study estimates in the paper use the earliest possible month (11 months before R01 expiry) as the omitted time category. However, the `did` package used to implement the [13] estimator requires there to be a pre-treatment period i.e. month -11 cannot be the omitted month. To make these results comparable with the main results, I let month -10

**Table 4. Count of employees paid by PI at one year before expiry.**

| | Occupation | 1 R01 | | | 2+ R01 | | |
|---|---|---|---|---|---|---|---|
| | | median | mean | sd | median | mean | sd |
| Count | All | 4.00 | 5.52 | 6.97 | 8.00 | 11.02 | 14.85 |
| | Faculty | 1.00 | 1.47 | 2.13 | 2.00 | 3.15 | 6.16 |
| | Postgraduate | 0.00 | 0.65 | 1.50 | 1.00 | 1.30 | 1.91 |
| | Research | 1.00 | 1.16 | 2.09 | 1.00 | 2.51 | 4.62 |
| | Clinical | 0.00 | 0.07 | 0.38 | 0.00 | 0.15 | 1.26 |
| | Graduate Student | 0.00 | 0.87 | 1.72 | 0.00 | 1.38 | 2.25 |
| | Instructional | 0.00 | 0.02 | 0.15 | 0.00 | 0.06 | 0.35 |
| | Other | 0.00 | 0.10 | 0.43 | 0.00 | 0.15 | 0.55 |
| | Other Staff | 0.00 | 0.02 | 0.32 | 0.00 | 0.02 | 0.27 |
| | Research Facilitation | 0.00 | 0.63 | 2.22 | 0.00 | 1.57 | 4.10 |
| | Technical Support | 0.00 | 0.15 | 0.86 | 0.00 | 0.28 | 1.09 |
| | Undergraduate | 0.00 | 0.37 | 1.17 | 0.00 | 0.46 | 1.83 |

be the omitted time category and then subtract $\hat{\beta}_{-11}$ from all the point estimates and confidence bounds, so that month -11 is effectively the reference month.

**Spending recovery after renewal.**   To see how quickly spending recovers after R01 renewal, I repeat the main analysis on spending. Whereas the original analysis is centered

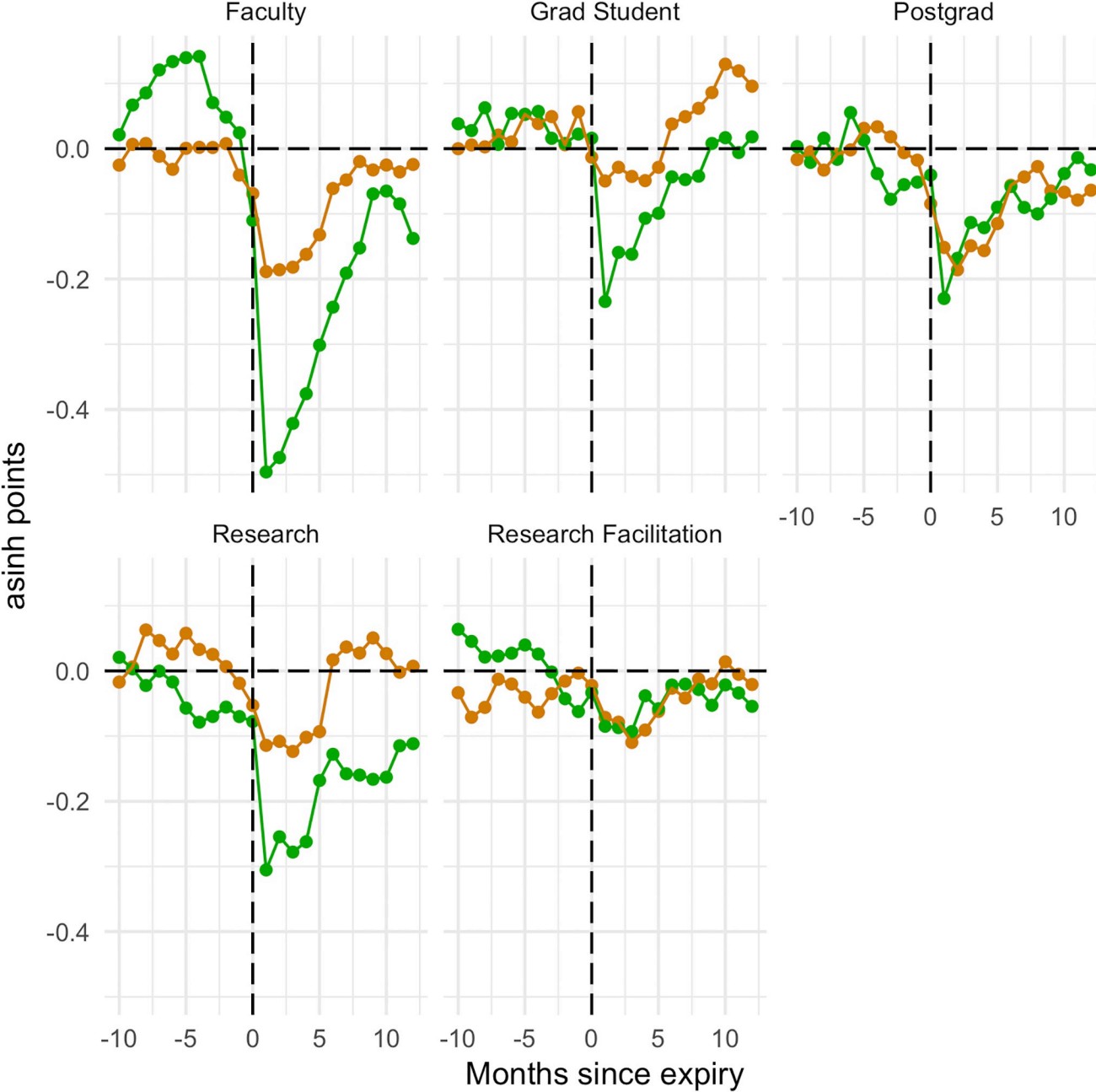

**Fig 20. This graph shows event-study estimates from a balanced panel of R01-PIs 12 months before and after the focal R01's expiry month, covering a period of 24 months.** The same specification is estimated for each occupation separately, where the outcome is the total number of employees of that occupation paid by the focal lab/PI. The regressions include R01-PI fixed effects and relative-to-expiry month fixed effects. Month 0 is the month that the project's budget expires. These regressions are run separately on subsamples of PIs that have one R01 grant (top) or multiple R01s (bottom), including R01-equivalents and P01 grants. Month -11 is the excluded category for the regression. 95% confidence intervals are clustered by expiring R01 project period. Percentage changes (plotted as text) are calculated using the median number of employees for interrupted labs at month -11 as baseline.

around the date of R01 expiry, this is centered around the date of R01 renewal and spans one year before and one year after renewal. The event study coefficients in Appendix Fig 18A show that once funds are available, there is a noticeable jump in spending. However, the recovery in spending is not immediate and takes about three months. The pattern is similar for labor payments (Appendix Fig 18B), while vendor payments recover more slowly ((Appendix Fig 18C)), which is also the case in the analysis in the main text.

Fig 19A to 19C repeat the same exercise while allowing for the event study coefficients to vary by interruption length. Total spending and labor payments do not recover differently for PIs that experienced longer interruptions. However, vendor payments recover faster for longer interruptions.

### Employee counts at PI/Lab-level

Table 4 displays summary statistics on the employees paid by a PI one year before R01 expiry, for the sample of PI-R01-renewals used in the main analysis of the paper.

**Event studies of employee counts by occupation.**   Fig 20 repeats the same analysis but using counts within occupation. I show the results for the five most common occupations: faculty, postgraduate researchers, graduate students, research, and research facilitation. Except for Research Facilitation, we see a similar pattern for all categories as we do for the total employee count.

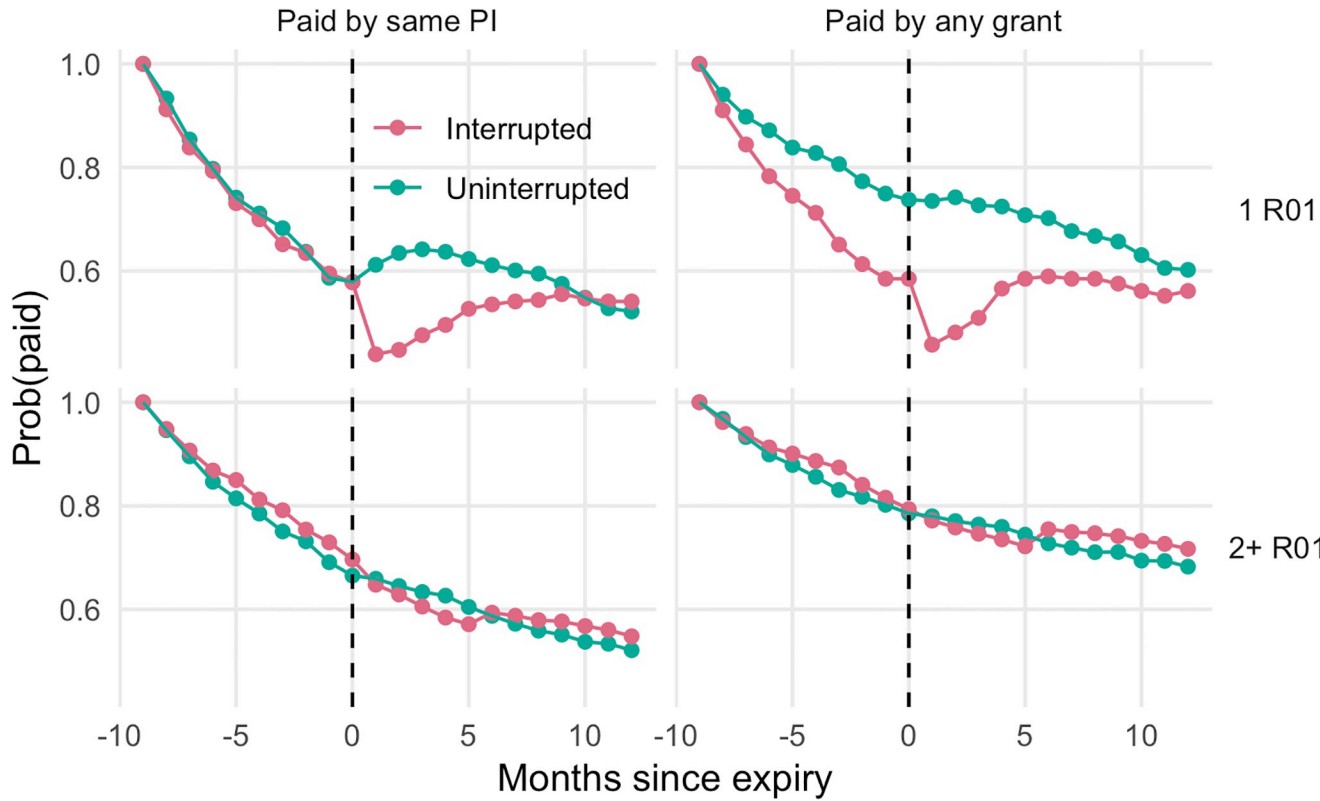

**Fig 21. The left column of this figure plots the average probability every month that an employee is paid by the focal PI.** The right column plots the average probability that an employee is paid by any grant at all. Employees linked to one R01 are represented in the top row. Employees linked to 2 or more R01s are represented in the bottom row.

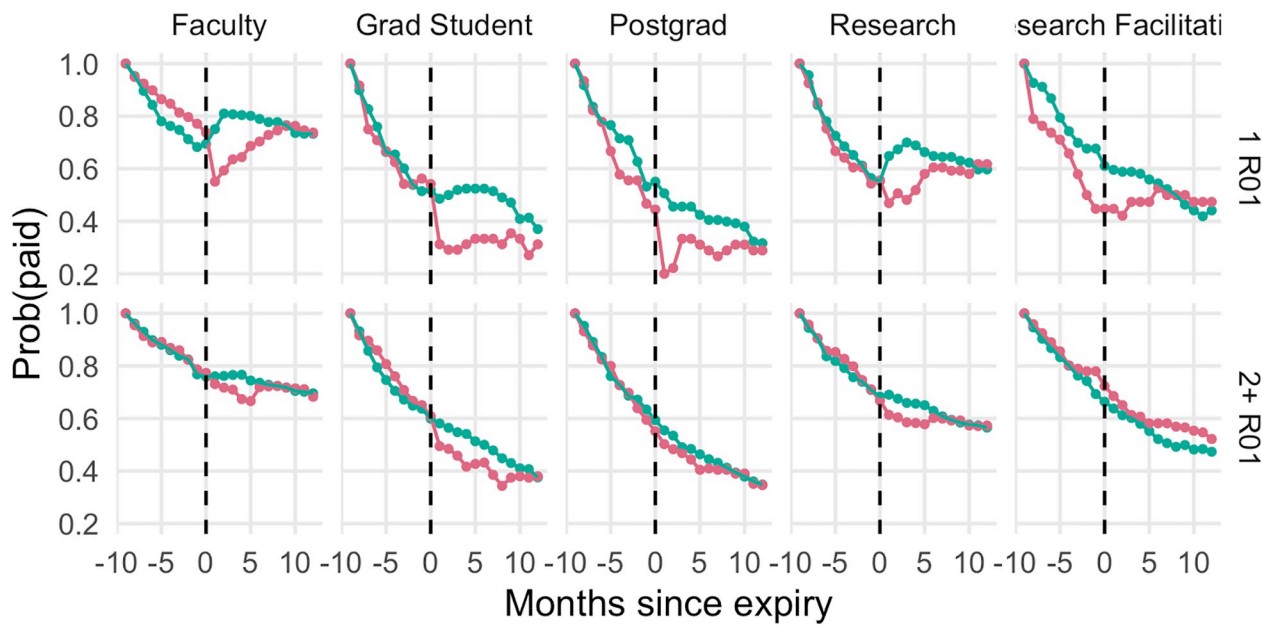

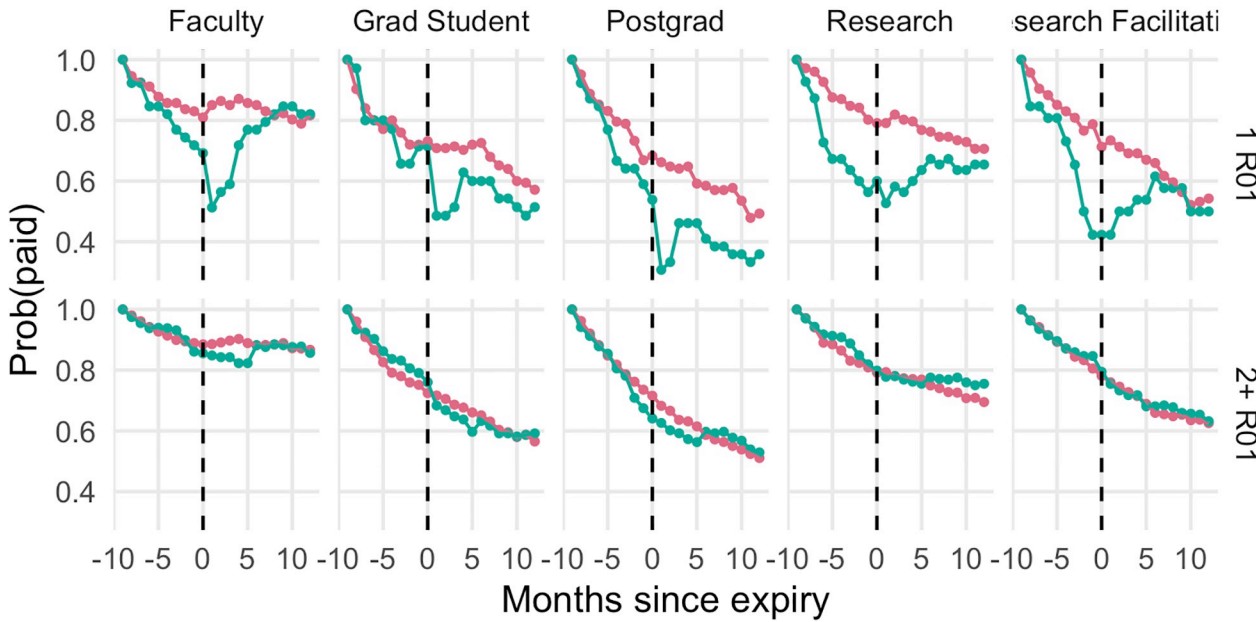

**Fig 22. This figure plots the average probability of being paid by the same PI or any grants at all in a given month for employees on interrupted (green) and uninterrupted projects (red).** The data are also subset by employees associated with only one R01-equivalent or 2 or more R01-equivalents.

## Employee-level results

Fig 21 shows the average probability each month of being paid by the same PI and being paid by any grant at all for employees associated with interrupted and uninterrupted R01s. In both cases, the probability of being paid diverges between interrupted and uninterrupted employees. This divergence begins earlier for the "any grant" outcome. Fig 22 repeats this analysis by employee occupation.

## Publications

**Coarsened exact matching.**   To estimate the effect of interruptions on publications, I find all instances where an R01 was successfully renewed within the fiscal year it expires. I then stack all combinations of renewed R01s and PIs of those R01s to create an R01-PI panel.

For PI characteristics, I use Author-ity [11], a dataset of disambiguated author names based on a snapshot of MEDLINE in 2009, and which has been probabilistically linked to PI IDs in ExPorter through the AuthorLink dataset.

I apply Coarsened Exact Matching [20]. The variables I match on are: gender, career age at the time of R01 expiry, and publications (raw counts and weighted by 3-year forward citations) in the pre-treatment period (before R01 expiry). Career age is coarsened at 10-year intervals. Pre-treatment publications are coarsened at percentiles 0, 25, 50, 75, 90, 95. I then estimate event study specifications, with the estimates plotted in Fig 23.

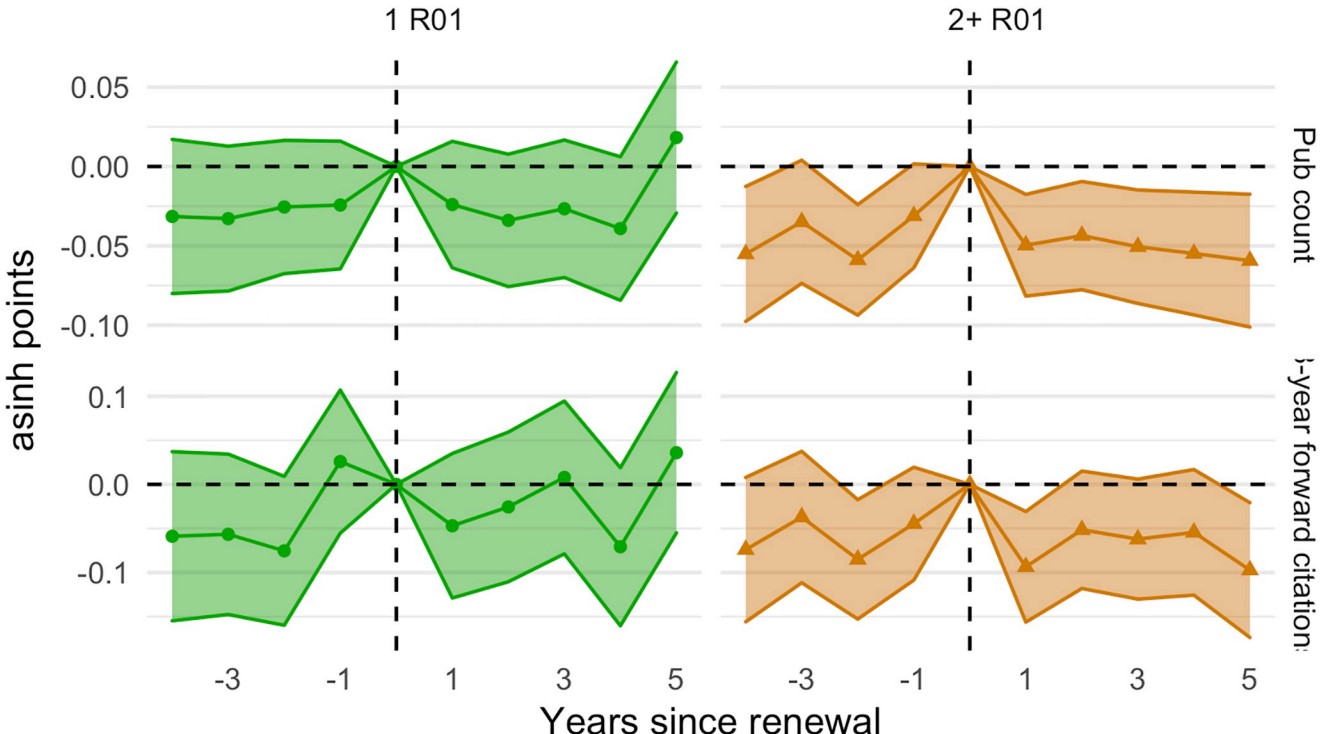

**Fig 23. This figure plots the event study coefficients estimating the difference in publication counts (arcsinh-transformed) between PIs that had an interrupted R01 and PIs that had a continuously funded R01, relative to publications in the year of R01 renewal.** R01-PI and treatment cohort-calendar year fixed effects are included. 95% confidence intervals are clustered by expiring R01 project period. The left/red plot is for PIs that only had one R01 and the right/blue plot is for PIs that had equivalent grants.

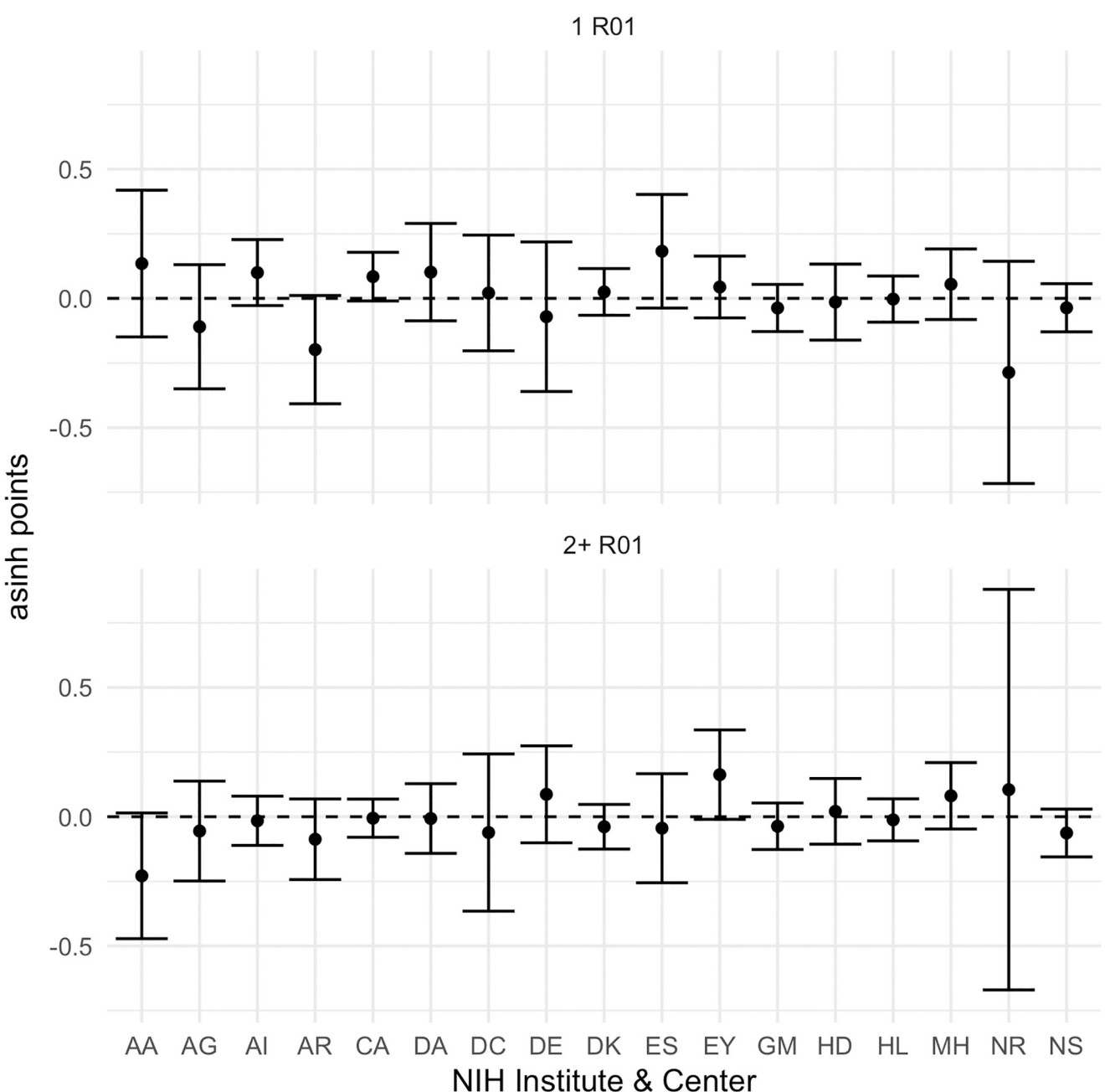

**Fig 24. This figure shows "static" difference-in-difference estimates and 95% confidence intervals of the difference in publication output if a PI had an interrupted R01, estimated separately on NIH IC subsamples.** The regression includes treatment-cohort-by-year and PI-R01-renewal fixed effects. Dependent variables are raw publication counts, arcsinh-transformed. Standard errors clustered by expiring R01 project period.

**Event study.** *Heterogeneity*. I repeat the static difference-in-difference estimates on sub-samples by NIH IC (Fig 24) and by whether a PI was above or below the median career age (Fig 25). The estimates are statistically insignificant in all cases and overall there is no indica-tion of a detectable effect of interruptions on research output.

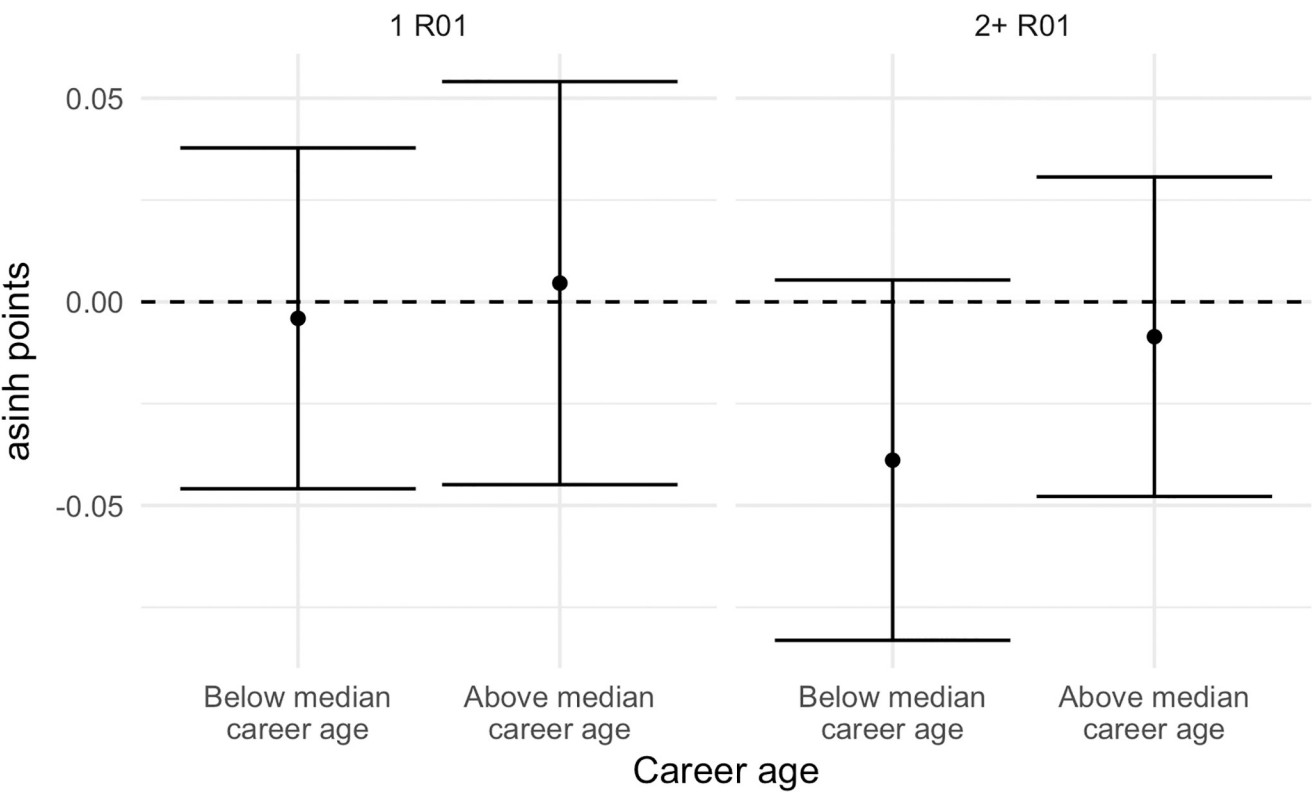

**Fig 25. This figure shows "static" difference-in-difference estimates and 95% confidence intervals of the difference in publication output if a PI had an interrupted R01, estimated separately on subsamples divided by whether the PI was above or below the median career age in the sample.** The regression includes treatment-cohort-by-year and PI-R01-renewal fixed effects. Dependent variables are raw publication counts, arcsinh-transformed. Standard errors clustered by expiring R01 project period.

## Supporting information

**S1 File.**
(PDF)

## Acknowledgments

I am grateful to Bruce Weinberg, David Blau, and Kurt Lavetti, for their invaluable guidance when I started this project as a graduate student. I am thankful to Kyle Myers for his feedback and support in the later stages of this project. This work would not have been possible without the support of IRIS at the University of Michigan, with special thanks to Natsuko Nicholls and Beth Uberseder. I am grateful to BriAnne Crowley for sharing her time and insights on grant management. The paper benefitted from helpful comments by discussants and participants at: the Aug 2018 UMETRICS meeting, Census ARiS Brown Bag, NBER-IFS International Network on the Value of Medical Research meeting, OSU Micro Lunch, SUNY Albany Department of Economics Seminar, MTEI Seminar at EPFL, RISE2 Workshop at the Max Planck Institute for Innovation and Competition (MPI), the Economics of Science & Engineering Seminar at Harvard Business School, and the 14th Workshop on the Organisation, Economics, and Policy of Scientific Research at MPI.

## Author Contributions

**Conceptualization:** Wei Yang Tham.

**Data curation:** Wei Yang Tham.

**Formal analysis:** Wei Yang Tham.

**Investigation:** Wei Yang Tham.

**Methodology:** Wei Yang Tham.

**Project administration:** Wei Yang Tham.

**Software:** Wei Yang Tham.

**Validation:** Wei Yang Tham.

**Visualization:** Wei Yang Tham.

**Writing – original draft:** Wei Yang Tham.

**Writing – review & editing:** Wei Yang Tham.

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
