## [Decision Letter · Decision Letter 0]

14 Jun 2022

PONE-D-22-12290Science, Interrupted: Funding Delays Reduce Research Activity but Having More Grants HelpsPLOS ONE

Dear Dr. Tham,

Thank you for submitting your manuscript to PLOS ONE. After careful consideration, we feel that it has merit but does not fully meet PLOS ONE’s publication criteria as it currently stands. Therefore, we invite you to submit a revised version of the manuscript that addresses the points raised during the review process. Both reviewers are quite positive about the contribution your analysis makes and think the paper has the potential to be even better.  In their comments they note a number of questions that are raised but not fully addressed in the current version.  In addition, they make a number of suggestions concerning the empirical implementation.  In particular around the choice of control group, potential for selection into treatment and the use of event study methodology.  Finally, they offer some useful suggestions for improving the clarity of presentation.

I encourage you to consider adopting the changes recommended by the reviewers, or to clarify the reasons for the choices you have made where you diverge from them.

We look forward to receiving your revised manuscript.

Kind regards,

Joshua L Rosenbloom

Academic Editor

PLOS ONE

Journal Requirements:

5. Please ensure that you refer to Figure 13 in your text as, if accepted, production will need this reference to link the reader to the figure.

Reviewers' comments:

Reviewer's Responses to Questions

**Comments to the Author**

1. Is the manuscript technically sound, and do the data support the conclusions?

Reviewer #1: Yes

Reviewer #2: Yes

2. Has the statistical analysis been performed appropriately and rigorously? 

Reviewer #1: Yes

Reviewer #2: Yes

3. Have the authors made all data underlying the findings in their manuscript fully available?

Reviewer #1: No

Reviewer #2: Yes

4. Is the manuscript presented in an intelligible fashion and written in standard English?

Reviewer #1: Yes

Reviewer #2: Yes

5. Review Comments to the Author

Reviewer #1: Report for: “Science, Interrupted: Funding Delays Reduce Research Activity but Having More Grants Helps”

The paper shows that when NIH grant funds are interrupted (delay in renewal), there is a reduction in expenditures by labs inputs and employees. This reduction is limited to PIs who have no other R01 grants that may be used to mitigate the impacts.

The paper is well written, interesting and important. I commend the author for their analysis. I hope my suggestions below help improve any lingering issues:

1. Temporary nature, and aggregate effect: The effects seem temporary; but when the expenditures rebound, the do not rise high enough to compensate. Can we quantify, the overall change in expenditures (i.e., the stock, not flow) over the two years, starting 3 months before the interruption? This would help us get a take-home number on the long-run reduction in spending by the lab because of the renewal delay.

2. Event-studies literature: There is a long new literature on event-studies specifications. The author has cited the literature (Goodman-Bacon 2018; Callaway and Sant’Anna 2018; Abraham and Sun 2018; Borusyak and Jaravel 2017), but it would be great to see the methods applied. The code is readily available on their websites, so implementation isn’t too difficult. The author also says “The Online Appendix discusses these in detail”, but I had trouble finding it there. It would be good to see the event studies using the methods cited (Callaway and Sant’Anna 2018; Abraham and Sun 2018; Borusyak and Jaravel 2017), or in fact using the method of the stacked-regression in the appendix of the Cenzig, Dube, Lindner, Zipperer (2019) QJE paper.

3. Three months pre-trends and selection into delay: The fall in expenditures begins 3 months before the interruption. Can we have a short discussion about why 3 months before? The reader may be worried that this is a sign of pre-trends, and so “selection into treatment” – that is, the projects that were doing poorly, and so cutting expenditures, were the ones that were not renewed. So we may have reverse causality: the delay doesn’t lead to a reduction in expenditures, but rather the project doing poorly causes the delay. Any way to help understand why this concern is not relevant would help.

4. Minor: The first regression equation has \\delta_{LR} fixed effects, but the text following it discusses it as \\delta_{iR} (employee level instead of PI level). I believe it’s just a minor typo.

5. Minor: Online Appendix: Many times the text refers the reader to the Appendix, but it is hard to find where in the appendix we should look at. I suggest saying “refer to Online Appendix Section A.3” or “Figure A.4 in the Online Appendix”.

Reviewer #2: The paper submitted analyzes the impact of delays in R01 grant recipients receiving their funds on both the employment of individuals in their labs and the scientific output of the labs. The paper finds that delays in funding has a sharp, but temporary adverse effect on employment for PIs with a single R01. In contrast, PIs who have multiple R01 grants show little change in their employment of graduate students, undergraduates, and research facilitators. Additionally, while these funding delays have large, temporary effects on employment, they do not significantly decrease the rate of production of publications.

Overall, I enjoyed reading the paper, find it methodologically sound, and consider the finding of a sizeable impact of funding delays to be compelling. Below I list a few questions and suggestions for revising the manuscript.

1. The analysis is well done and compelling. The missing component, however, is what the practical impact is of the large decline in both working employees and vendor spending. Take the employees, for example. If graduate students are released from labs for some number of months, are they truly not working? If so, how does that impact the career success of those graduate students? Are they more likely to not graduate if they had that disruption in their work? Or, do graduate students continue to work in the labs but without compensation—creating the dip you find in your analysis, but without an impact on the training of the graduate students themselves? Figure 4 begins to address this, but teasing out which of these stories is correct is important since in some the impact is fairly limited while in others the impact is substantial and would be a call for additional policy response.

2. The empirics are generally clear and appropriate. The question that I would like the paper to address is why you used the control group that you chose for the analysis. One might imagine running similar analysis using the same labs but during an earlier time period as a control. You could also imagine restricting the control group to be within the same university or field. I would suggest that the motivation for the choice of control group and the reason for dismissing other potentially valid control group specifications be made explicit in the paper.

3. It is not clear what the aggregate impact of funding delays is on science. If scientists use less of their funds when the funding is delayed, they will presumably be able to spend more later to make up for that dip and expend the entirety of their grant. In the event study plots, however, it is not clear where the labs are spending more than their steady-state level. Could you please explain why we don’t see higher than average spending after the dip is done? Could you also try to estimate what is the total impact over the course of the grant if labs spend less in the period when funding is delayed and more in later periods?

4. The most notable heterogeneity is the difference between labs with one versus multiple R01 grants. Could you provide additional information about why labs are able to compensate for delays using other R01s but are not able to compensate with other non-R01/departmental funds? Relatedly, could please provide a figure that shows that PIs with multiple R01s increase their spending on their other R01 to compensate for the decreased spending on the delayed R01 (if true). If you do not find a commensurate increase in expenditures on the unaffected second R01 grant then why are PIs with multiple grants able to maintain their level of spending?

5. Additional exploration of heterogeneity by fields, size of lab, PI age, and other attributes could be very interesting to readers. For example, Appendix Section 1.3 clearly shows heterogeneity in the rates of interruptions across fields. Similarly, heterogeneity in the impact on employees by employee attributes could be of interest to policymakers.

6. Appendix Section 1.2 says that the time series of interruptions provide evidence that the NIH is responsive to delays in the federal budgeting process. Could the author please expand on how the figure demonstrates this?

7. The author sometimes refers to labs. Can the author please clarify if a single PI is equivalent to a “lab” or if a lab can have multiple PIs?

8. Why is the preferred specification in the main text not matched on project period length and the matched version in Appendix 3.1.1 instead of the other way around? I find the matched project length version more compelling, however, if there is a reason to prefer the unmatched version, I would ask the author to note that in the empirical framework section.

9. What is the difference between the employees with the title “Research” and the title “Research Facilitation”?

6. PLOS authors have the option to publish the peer review history of their article (what does this mean?). If published, this will include your full peer review and any attached files.

Reviewer #1: **Yes: **Gaurav Khanna

Reviewer #2: No

---

## [Author Response · Author response to Decision Letter 0]

17 Oct 2022

Please see attached PDF labelled "Response to Reviewers"

---

## [Decision Letter · Decision Letter 1]

14 Nov 2022

PONE-D-22-12290R1Science, Interrupted: Funding Delays Reduce Research Activity but Having More Grants HelpsPLOS ONE

Dear Dr. Tham,

Thank you for submitting your manuscript to PLOS ONE. After careful consideration, we feel that it has merit but does not fully meet PLOS ONE’s publication criteria as it currently stands. Therefore, we invite you to submit a revised version of the manuscript that addresses the points raised during the review process.

Both reviewers state that you have addressed the bulk of the concerns raised regarding the initial submission.  This version is very close to ready for publication.  Reviewer #2 raises a number of questions, however, that should not require any further data analysis or investigation, but if answered will enhance the usefulness and impact of your article. I will not need to seek external review at this point. However, I want to give you the opportunity to strengthen your contribution by considering the queries from Reviewer #2. These are mostly asking for clarification of your choices or providing further clarification.  Simply summarize the changes you have made or your reasons for not making changes in the response to reviewers.  Once I am satisfied you have given these suggestions careful thought I will be happy to accept this article for publication.

We look forward to receiving your revised manuscript.

Kind regards,

Joshua L Rosenbloom

Academic Editor

PLOS ONE

Journal Requirements:

Reviewers' comments:

Reviewer's Responses to Questions

**Comments to the Author**

1. If the authors have adequately addressed your comments raised in a previous round of review and you feel that this manuscript is now acceptable for publication, you may indicate that here to bypass the “Comments to the Author” section, enter your conflict of interest statement in the “Confidential to Editor” section, and submit your "Accept" recommendation.

Reviewer #1: All comments have been addressed

Reviewer #2: (No Response)

2. Is the manuscript technically sound, and do the data support the conclusions?

Reviewer #1: Yes

Reviewer #2: Yes

3. Has the statistical analysis been performed appropriately and rigorously? 

Reviewer #1: Yes

Reviewer #2: Yes

4. Have the authors made all data underlying the findings in their manuscript fully available?

Reviewer #1: Yes

Reviewer #2: Yes

5. Is the manuscript presented in an intelligible fashion and written in standard English?

Reviewer #1: Yes

Reviewer #2: Yes

6. Review Comments to the Author

Reviewer #1: I congratulate the authors for a well executed manuscript. I believe it would make an important contribution to the literature. I encourage the authors to share all data and code if possible.

I look forward to other papers that examine what happens to grad students.

Reviewer #2: Thank you for the opportunity to read this revised manuscript. I am delighted with the changes that the authors have made. I am particularly pleased with the way in which the authors addressed the staggered difference-in-differences literature.

I do believe that there are three areas that the authors could improve before publication. First, the section describing the data could be clearer. In particular, I would like more details about the sample selection brought into the main text. Second, the most interesting aspects of this paper are the heterogeneity in the effect of interruptions across researchers. The authors have done some work on this front, but I think that this could greatly enhance the impact of the paper. Lastly, while the authors have made progress on addressing why the total spending of researchers is less, I still feel that this could be more pointedly addressed.

1. The main text data section could be a clearer on the sample timeframe. Specifically, am I correct that the data on employment goes from 2001 to 2018, but the research output is measured for 2001 to 2013? The main data section mentions the raw databases that are linked together to form the analytical dataset, but it would be helpful if the authors also described the main analytical dataset, including what time period is actually covered for the outcome variable and covariates in the main text.

2. Some of the most interesting results from this paper could be the heterogeneity across research and subfield of research. I would recommend bringing Appendix D.4.3 into the main text and expanding on what can be learned from it. The heterogenous effects on young researchers are particularly interesting; is this evidence of a Matthew Effect? I am less clear on what the conclusion is regarding the heterogeneity across NIH centers, but would appreciate if the authors could expand on it.

3. I am still a little confused about the aggregate spending impact. If the total spending change is 52% decline, where did the remaining money go? I understand that a researcher might forgo hiring a graduate student, and thus the researcher will have lower labor costs. But does that mean that the researchers are simply not spending down their full grants now? If data is not available to address this, I would look qualitative evidence or comments from PIs regarding how they handle the excess funds at the end of their grant.

4. Figure 2(b) – This graph is very useful for understanding if there is a sharp distinction between those with less than a 30 day and those with more than 30 day interruption. Could you please bin the days of interruption, such that we can see with more granularity the distribution under ~60 days?

5. I appreciate the inclusion of Figure 6 showing the rate of interruptions over time. My one concern is that the timeline ends at 2014, but you are using data through 2018. Even if you filtered to PIs with a year’s worth of monthly data following the disruption, I would expect this graph to go through 2017. Could you please either update that figure or explain why it is cut at 2014?

6. On page 13, you say that the coefficients of interest are beta-t, however, I think this might be a typo. In the equation, I only see a beta-m. I think that to be consistent, it might be useful to stick to beta-e.

7. For the equations on page 14, I think that there are missing summation terms on the interaction of the beta coefficients and the relative-time indicators.

8. Given the lack of impact on research outcomes and yet the massive impact on employee counts (particularly for single R01), is there some way to tell if the labs employees are simply switching to uncompensated work? For example, can you see if the names of the researchers on the publications from the labs change when the PIs reduce the employment during the interruption? If you could show that there was uncompensated work done, this would be a contribution to policy debates about grad student and postdoc compensation.

9. The authors use the UMETRICS data through 2018. I imagine that the reason for not expanding into more recent years is because the authors do not want to conflate funding disruptions with the COVID-19 disruptions. I would encourage the authors to mention the motivation for the sample time frame chosen in the main text.

7. PLOS authors have the option to publish the peer review history of their article (what does this mean?). If published, this will include your full peer review and any attached files.

Reviewer #1: **Yes: **Gaurav Khanna

Reviewer #2: No

---

## [Author Response · Author response to Decision Letter 1]

8 Dec 2022

Response to Reviewers is attached as a pdf file.

---

## [Editor Report · Decision Letter 2]

4 Jan 2023

Science, Interrupted: Funding Delays Reduce Research Activity but Having More Grants Helps

PONE-D-22-12290R2

Dear Dr. Tham,

We’re pleased to inform you that your manuscript has been judged scientifically suitable for publication and will be formally accepted for publication once it meets all outstanding technical requirements.

Kind regards,

Joshua L Rosenbloom

Academic Editor

PLOS ONE
---

## [Editor Report · Acceptance letter]

20 Feb 2023

PONE-D-22-12290R2 

Science, Interrupted: Funding Delays Reduce Research Activity but Having More Grants Helps 

Dear Dr. Tham:

I'm pleased to inform you that your manuscript has been deemed suitable for publication in PLOS ONE. Congratulations! Your manuscript is now with our production department. 

Kind regards, 

on behalf of

Dr. Joshua L Rosenbloom 

Academic Editor

PLOS ONE